# Evaluation of Patellar Groove Prostheses in Veterinary Medicine: Review of Technological Advances, Technical Aspects, and Quality Standards

**DOI:** 10.3390/ma18071652

**Published:** 2025-04-03

**Authors:** Mateusz Pawlik, Piotr Trębacz, Anna Barteczko, Aleksandra Kurkowska, Agata Piątek, Zbigniew Paszenda, Marcin Basiaga

**Affiliations:** 1Department of Biomaterials and Medical Devices Engineering, Faculty of Biomedical Engineering, Silesian University of Technology, 41-800 Zabrze, Poland; mateusz.pawlik@cabiomede.com (M.P.); aleksandra.kurkowska@caabiomede.com (A.K.); apiatek80@gmail.com (A.P.); zbigniew.paszenda@polsl.pl (Z.P.); marcin.basiaga@polsl.pl (M.B.); 2CABIOMEDE Ltd., 25-663 Kielce, Poland; anna.barteczko@cabiomede.com; 3Department of Small Animal Surgery and Anaesthesiology, Institute of Veterinary Medicine, Warsaw University of Life Sciences-SGGW, 02-787 Warsaw, Poland

**Keywords:** additive manufacturing, patellar luxation, dogs, cats, patellar groove replacement

## Abstract

This review explores the technological advancements in, engineering considerations regarding, and quality standards of veterinary patellar groove replacement implants. Veterinary-specific regulations for these implants are currently lacking. Therefore, human knee implant benchmarks are used as references. These benchmarks guide evaluation of the surface quality, material selection, biocompatibility, and mechanical performance of the implant to ensure reliability and longevity. Patellar luxation is a common orthopedic disorder in small animals which leads to patellofemoral joint instability and cartilage degeneration, and is often caused by angular limb deformities that disrupt patellar alignment. In severe cases, patellar groove replacement is necessary to restore function and alleviate pain. The implant materials must provide durability, mechanical strength, and biocompatibility to withstand joint forces while ensuring minimal wear. High-quality surface finishes reduce the friction experienced by these materials, improving their long-term performance. Advances in 3D printing allow the creation of patient-specific implants. These implants offer an enhanced anatomical fit and enhanced functionality, which is especially beneficial in complex cases. However, challenges remain in achieving consistent manufacturing quality and economic feasibility. While custom implants are invaluable for difficult cases, standardized designs are sufficient for routine applications. Combining human implant standards with new manufacturing technologies improves veterinary orthopedic solutions. This integration expands the treatment options for patellar luxation and enhances the quality and accessibility of implants.

## 1. Introduction

### 1.1. 3D-Printed Patient-Specific Implants in Veterinary Medicine

The application of patient-specific 3D-printed implants in veterinary orthopedics has gained significant traction, particularly in complex reconstructive surgeries where off-the-shelf implants fail to provide an adequate anatomical fit and biomechanical stability. Several case reports and clinical studies have demonstrated the feasibility and benefits of customized implants for limb-sparing procedures and joint reconstruction in dogs.

Custom 3D-printed titanium plates and osteotomy guides have been successfully utilized for treating antebrachial limb deformities, allowing precise bone alignment and enhanced fixation stability. Similarly, patient-specific 3D-printed implants have been applied in cranial cruciate ligament treatment, offering tailored solutions that reduce the frequency of complications associated with conventional implants [1,2].

Notably, limb-sparing endoprostheses that incorporate 3D-printed titanium constructs have been used in dogs suffering from osteosarcoma, demonstrating promising functional outcomes and high levels of integration. The use of personalized porous titanium endoprostheses for reconstructing mandibular, radial, and tibial defects after tumor excision in canine patients has further validated the adaptability of this technology. Additionally, in the case of bilateral patellar groove replacement, a custom implant was shown to have successfully restored joint function and mitigated trochlear groove damage, highlighting the advantages of individualized design in addressing severe pathologies of the femoro-patellar joint [3,4,5,6].

While patient-specific implants are still emerging in veterinary medicine, 3D printing technology and surgical planning advancements have significantly improved the precision and fit of implants and improved long-term outcomes [7].

### 1.2. Patellar Groove Implants

The patellar groove, also known as the trochlear groove, plays a critical role in the biomechanics of both canine and feline ambulation. It is integral to the proper tracking and function of the patella, which in turn is essential for normal limb movement and stability [8]. Abnormalities seen in the patellar groove include hypoplasia or erosion. These abnormalities can result in conditions like medial patellar luxation (MPL), a frequent cause of lameness in dogs and cats [9]. The depth and morphology of the patellar groove vary significantly among different breeds. This indicates that breed-specific considerations are essential for the accurate assessment of patellar health and effective surgical planning [10,11]. Moreover, the successful surgical management of patellar luxation often involves procedures that restore or reconstruct the patellar groove, as evidenced by the frequency of trochleoplasty and patellar groove replacement [6,8,9,12,13,14]. These interventions aim to reestablish the normal alignment of the patellar mechanism and reduce the possibility of luxation. In summary, the patellar groove is a crucial anatomical feature in canine and feline orthopedics, and its integrity is necessary for maintaining normal stifle joint function. The accurate assessment of the patellar groove’s morphology and precise measurements obtained using advanced diagnostic methods like computed tomography are essential for effective treatment planning [10,11,15,16].

Given the lack of veterinary-specific standards, this study aims to review materials, manufacturing methods, and quality requirements for veterinary patellar groove replacement implants, a form of partial implant, as well as requirements and standards from human medicine. By conducting a thorough analysis of various sources, including recent empirical studies, clinical reports, technological advancements in implant manufacturing, and human medicine legal requirements and quality standards, this article aims to achieve the following specific goals:Explain the biomechanical and anatomical factors and pathologies that necessitate patellar groove replacement, thereby enhancing the understanding of the conditions that warrant surgical intervention;Review quality requirements and standards for the human medicine equivalent to veterinary patellar groove replacement, product—knee implants;Review the materials used to fabricate knee and patellar groove implants, concentrating on their mechanical properties, biocompatibility, surface modifications, and clinical performance, to inform future material selection;Compare the advantages and disadvantages of patellar groove implants available on the market;Highlight technological advancements, particularly in designing and manufacturing custom implants and surgical guides.

This review aims to provide veterinary orthopedic surgeons, researchers, and practitioners with a comprehensive resource that supports clinical decision-making and promotes ongoing innovation in the treatment of patellar groove disorders in companion animals. Referencing human medical standards, particularly those related to the design, materials, manufacturing methods, and quality requirements of knee implants, this review offers detailed insights into how veterinary implants should be designed and manufactured. By following these guidelines, veterinary implants can achieve the necessary safety and quality levels, ultimately enhancing surgical outcomes and ensuring the well-being of animal patients.

### 1.3. Literature Selection Methodology

In preparing this review, a comprehensive search of the available literature was conducted to evaluate the current knowledge of patellar groove implants, particularly in veterinary applications. Due to the highly specialized nature of this topic, the literature pool is inherently limited, with only four veterinary studies addressing patellar groove replacement being identified. These include three studies focusing on off-the-shelf implants and one case of a custom-designed solution. These studies highlight different clinical scenarios where patellar groove replacement has been applied, ranging from severe osteoarthritis and trochlear groove erosion to cases requiring patient-specific implants due to anatomical abnormalities or iatrogenic trochlear damage. Given the scarcity of available research, these studies serve as the primary reference points for understanding the application of patellar groove replacement in veterinary orthopedics [6,17,18,19].

Given the scarcity of veterinary-specific research, this review also draws upon established principles from human orthopedic implantology, where extensive studies exist on knee prosthetics and surface quality optimization. The sources regarding human implantology were selected based on their relevance to implant materials, biomechanics, and surgical applications, and included peer-reviewed journal articles, clinical case reports, and standards relevant to implant design. The search included databases such as PubMed, Scopus, and Web of Science, and used keywords such as “patellar groove implant”, “trochlear prosthesis”, “veterinary knee prosthesis”, and “custom orthopedic implants”.

Since no formal systematic review methodology was followed, this review aims to synthesize available knowledge to highlight the clinical and technological gaps in the area of veterinary patellar groove replacement. Human medical standards provide a framework for understanding the potential advancements in and limitations of veterinary applications, emphasizing the need for further dedicated research in this field.

## 2. Anatomy and Physiology

The distal epiphysis of the femur consists of two condyles, each topped by an epicondyle. The fossa intercondylaris separates the two condyles.

The trochlear groove (the patellar groove) is formed between the trochlear ridges at the cranial aspect of the femur, where the patella acts as a pulley. The patella is an ossification (sesamoid bone) within the insertion of the quadriceps femoris muscle tendon, and improves the efficiency of the extensor mechanism [20] (Figure 1).

The tendon between the patella and its insertion on the tibial tuberosity is referred to as the patellar ligament. In the caudal aspect of the stifle joint, two additional sesamoid bones are present at the fabellae at the end of the gastrocnemius muscle [20].

The anatomy and physiology of the patellar groove are fundamental to the function of the stifle joint in dogs and cats. The morphology of this groove, including its depth and shape, is critical for regular patellar tracking and joint stability. Abnormalities in the morphology of the trochlear groove can lead to issues such as patellar luxation, which significantly affects joint stability and necessitates surgical intervention [10,11,15,16].

### 2.1. The Main Causes of Patellar Dislocation and Indications for Patellar Groove Arthroplasty

Patellar luxation is a multifaceted orthopedic condition primarily resulting from angular deformations of the femur and tibia. These structural abnormalities disrupt the alignment of the quadriceps mechanism, leading to patellar displacement from the trochlear groove. While secondary changes in the trochlea, such as a shallow groove, are frequently observed, they are generally consequences of the underlying malalignment [21,22,23].

#### 2.1.1. Role of Angular Deformities in Patellar Luxation

Angular deformities are the predominant contributors to patellar luxation, disrupting the biomechanical forces that maintain patellar alignment. Commonly observed deviations include:Lateral Bowing of the Distal Femur (Varus Deformity)

This deformity shifts the trochlear groove laterally, increasing the likelihood of medial patellar luxation;

Hypoplasia of the Medial Condyle

A reduced medial condyle provides inadequate support to the patella, exacerbating existing instability;

Torsion of the Tibial Tuberosity

This torsion misaligns the pull of the quadriceps mechanism, increasing medial or lateral luxation risks;

Medial Bowing of the Proximal Tibia

This deformity further shifts forces medially, contributing to patellar displacement.

These deviations lead to chronic instability and force the patella to traverse abnormal paths, which can accelerate the wear and degeneration of the joint [21,22,23].

#### 2.1.2. Secondary Changes in the Trochlea

The absence of the patella from the trochlear groove during critical developmental periods often results in insufficient sulcus formation. This secondary condition exacerbates existing instability and increases the severity of luxation in affected joints. However, these changes are typically a consequence of the primary malalignment rather than an independent cause [21].

#### 2.1.3. Indications for Patellar Groove Arthroplasty

Patellar groove replacement (PGR) is a method of arthroplasty used to treat severe patello-femoral osteoarthritis and severe patello-femoral instability, which is often associated with patellar luxation [6,17]. Patellar luxation is a common pathology necessitating patellar groove replacement in canine patients. It is characterized by the displacement of the patella from the trochlear groove, with medial luxation being more prevalent than lateral luxation, particularly in small-breed dogs and females [24]. Patellar luxation is often linked to a shallow trochlear groove, with the severity of luxation correlating with the degree of trochlear dysplasia [9,10,15]. This condition can lead to severe patellofemoral osteoarthritis, as evidenced by the cartilage erosion that is often seen on the articular surface of the patella, especially in higher grades of luxation and heavier dogs [25] (Figure 2).

Trauma, such as a distal femur fracture, may also necessitate PGR, as seen in a case where a Staffordshire bull terrier required internal fixation following PGR surgery [26]. Additionally, degenerative diseases that result in the wearing out of the medial and lateral trochlear ridges can lead to the need for PGR, as demonstrated in a Toy Poodle with bilateral medial patellar luxation [6]. Interestingly, while PGR is a relatively novel method, it has been used successfully in cases where previous surgical interventions failed to stabilize the patella [17]. Moreover, the presence of patella alta, a condition defined as a patellar ligament length to patellar length ratio greater than two, has been associated with medial patellar luxation and may influence the decision to perform PGR [27].

Advanced imaging techniques, including computed tomography (CT) and magnetic resonance imaging (MRI), are instrumental in assessing the morphology of the trochlear groove. These technologies provide detailed insights that are crucial for surgical planning and evaluating the extent of patellar luxation [10,11,16,28].

## 3. Standards and Requirements for Knee Implants in Human Medicine: A Reference for Veterinary Applications

In contrast to human medicine, where implants are subject to strict regulatory standards, veterinary implants are not governed by formal legal requirements. Nevertheless, due to the considerable biomechanical and anatomical similarities between the human knee and the canine stifle joint, we propose that human implant standards, should be analyzed and compared for their potential applicability in veterinary medicine. This paragraph examines the justification for such comparisons by drawing upon existing research on joint biomechanics and associated pathologies.

While the specific forces in the human knee and canine stifle joint are not explicitly detailed in the current literature, relevant insights can be drawn from broader studies on joint biomechanics and pathologies.

Irvine [29] discusses the impact of knee pressure variations on joint stability and cartilage nourishment, emphasizing the role of dynamic forces during movement. This is pertinent to humans and dogs, as knee stability and cartilage health are crucial for proper joint function and mobility in both. Similarly, Murakami [30] provides an overview of the loads experienced by human joints, including the knee, and highlights the importance of kinematics and alignment in maintaining joint function. These principles also apply to canine joints, where proper alignment and load distribution are among the components of the biomechanical system and are essential for preventing injuries and degenerative conditions.

Significant biomechanical similarities exist between the human knee and the canine stifle joint. Therefore, adopting human medical standards as a reference for veterinary implants is a logical and practical approach. These implants essentially operate under the same biomechanical principles of the implant–living tissue interface, whether in humans or animals.

ISO standards in human medicine, including implants, ensure medical device safety, reliability, and efficacy, safeguarding patient health through the application of strict requirements for materials, manufacturing processes, and quality control. These standards embody an ethical commitment to the well-being of living beings, promoting the development of implants that minimize risks, enhance patient outcomes, and uphold the highest standards of care and responsibility in medical practice.

ISO standards for human knee implants, such as those addressing their material properties, mechanical performance, and biocompatibility, provide a robust framework that is suitable for veterinary applications for which there are no specific standards in the veterinary field. By leveraging these established guidelines, veterinary implants can be designed to meet animal patients’ unique anatomical and physiological needs while ensuring high levels of safety and effectiveness.

In conclusion, applying human knee implant standards in veterinary medicine is logical, given the absence of specific regulations. The shared aspects of joint biomechanics and pathologies between humans and dogs support adapting these established standards to veterinary applications. By utilizing these guidelines, veterinary practitioners can ensure that implants are designed and manufactured to meet animal patients’ unique anatomical and physiological needs. This approach enhances surgical outcomes and promotes the ethical responsibility of providing safe and effective care to animal companions. Further research to explore biomechanical parallels between the human knee and canine stifle joint, which would refine the application of these standards and validate their suitability for veterinary orthopedics, is encouraged.

In conclusion, using human knee implant standards in veterinary medicine is a practical and ethical solution to the lack of specific regulations. These standards, addressing critical aspects such as material properties, mechanical performance, and biocompatibility, should be adopted by implant manufacturers to ensure that their products meet rigorous quality requirements. By adhering to these guidelines, manufacturers can provide safe, reliable implants that are suitable for animal patients’ unique anatomical and physiological needs.

Additionally, these standards offer a valuable tool for veterinary practitioners to evaluate the quality of implants. Surgeons can prioritize products that comply with recognized ISO standards over those that lack clear information about their quality and adherence to established guidelines. This empowers practitioners to make informed decisions and promotes the use of high-quality implants, enhancing surgical outcomes and patient care.

Adopting these standards would foster consistency in manufacturing and usage while ensuring ethical responsibility in providing safe and effective treatment for animal companions. Further research into the biomechanical similarities between the human knee and the canine stifle joint will refine and validate the application of these standards, advancing the field of veterinary orthopedics and supporting the delivery of superior care to animal patients.

### 3.1. ISO Standards for Orthopedic Implants

ISO standards are pivotal in ensuring the safety, biocompatibility, and mechanical performance of orthopedic implants by defining stringent requirements for materials, manufacturing processes, and quality management systems. These standards are essential for developing and producing human implants and provide a suitable framework for veterinary implant manufacturing, which does not have dedicated regulations.

Top-quality manufacturers of veterinary implants frequently adhere to critical ISO standards to maintain product safety and reliability. For example, ISO 5832 specifies the material properties required for surgical implants, ensuring biocompatibility and mechanical integrity. Additionally, ISO 13485 outlines comprehensive quality management systems for medical devices, addressing all product lifecycle stages, including design, development, manufacturing, inspection, testing, packaging, and documentation. Veterinary implants developed under these internationally recognized standards offer superior reliability and consistent performance, ultimately improving surgical outcomes and patient care (Table 1) [31,32,33,34,35,36,37,38,39,40,41,42,43,44,45].

Practitioners can use adherence to these ISO standards as a benchmark for evaluating the quality of veterinary implants. The development of products under certified standards provide greater assurance of their safety and efficacy, allowing surgeons to make informed decisions when selecting implants for their patients. Conversely, the absence of such certifications raises concerns about product reliability, as adherence to recognized standards is critical for ensuring consistent quality and minimizing risks during clinical application.

By highlighting the importance of ISO standards in veterinary implant manufacturing, this discussion underscores the necessity of selecting products that meet these rigorous benchmarks, fostering improved outcomes and enhanced patient care in veterinary orthopedics.

### 3.2. Comparing Human and Veterinary Knee Biomechanics: Implications for Implant Design

While human knee prostheses provide an essential benchmark for veterinary implants, quadrupedal biomechanics differ significantly from bipedal locomotion, requiring species-specific design considerations. Force plate analysis and finite element modeling studies have demonstrated that:The peak joint forces in the canine stifle reach 3.5–4.5 times the animal’s body weight during high-impact activities such as running or jumping. In comparison, human knee prostheses are designed to withstand 5–7 times the human body weight, particularly during stair climbing or squatting;In dogs, the tibiofemoral contact areas are smaller relative to their body weight, resulting in higher localized pressures on articulating surfaces;Human knees achieve full extension, while canine and feline stifles remain in a mild flexion, influencing implant kinematics and stress distribution [46]

These differences suggest that, while human standards serve as a valid reference, veterinary implants must be adapted in terms of material selection, implant geometry, and load distribution characteristics to function optimally under quadrupedal locomotion.

### 3.3. Critical Aspects of Joint Prosthesis

Although patellar groove replacement implants are not used in human medicine, the requirements for surface quality, materials, biocompatibility, and mechanical performance align closely with those of human knee implants, making them the most appropriate reference point.

Developing such implants requires meticulous attention to three critical aspects: their surface roughness, their biocompatibility, and maintaining stringent cleanliness standards. The surface roughness is paramount, as it directly influences the implant’s wear resistance, articulation, and impact on surrounding tissues. Biocompatibility ensures that the materials, such as titanium alloys, are safe and suitable for long-term implantation. Finally, stringent cleanliness standards during manufacturing are vital to minimizing the risk of postoperative complications, such as infections, which can compromise the implant’s success.

By leveraging the established benchmarks and rigorous standards for human knee implants, veterinary patellar groove replacement implants can achieve the high levels of quality and performance necessary for optimal clinical outcomes despite the absence of dedicated standards in veterinary medicine.

### 3.4. Joint Articulating Surfaces—Roughness

Surface roughness is a crucial factor in the performance of knee joint prostheses. The articulating surfaces of these implants must be smooth to minimize friction and wear, which is essential for the implant’s longevity and the patient’s comfort. ISO 7207-2 specifies the surface roughness requirements for the articulating surfaces of partial and total knee joint prostheses. According to this standard, all articulating surfaces of a metallic or ceramic femoral component must have a roughness Ra_max value of ≤0.1 μm (very high gloss mirror finish). Measurements should be taken across the articulating surface in an approximately square grid, with locations no more than 10 mm apart (Figure 3) [39,40].

Applying these standards directly to veterinary implants presents challenges due to animal prostheses’ significantly smaller size than human implants—the entire articulating surface is often smaller than the measurement grid itself. This discrepancy implies that the same grid spacing would provide insufficient data for veterinary implants, as fewer measurement points would be captured over the smaller surface area.

To address this limitation, a more densely spaced measurement grid must be used to comprehensively evaluate the surface roughness of smaller veterinary implants. Thus, while human standards serve as a valuable starting point, they must be adapted to accommodate the unique size constraints of veterinary implants, ensuring precise measurements and maintaining the high standards that are essential for minimizing wear and ensuring the longevity and functionality of veterinary implants.

### 3.5. Materials Biocompatibility

Biocompatibility is paramount in the selection of materials for knee joint prostheses. The materials must not induce any adverse reactions in the body, such as inflammation or allergic responses. The ISO 10993 series provides comprehensive guidelines for the biological evaluation of medical devices, including cytotoxicity, sensitization, and systemic toxicity tests. Adherence to these standards ensures that the materials used in prostheses are safe for long-term implantation [36,37,38].

Certified and ISO-standardized materials, such as those specified in the ISO 5832 series, are essential for constructing knee joint implants. Using medical-certified materials guarantees biocompatibility and mechanical properties that can withstand the physiological loads encountered in the knee joint. This is particularly important for custom solutions where the patient’s needs might necessitate unique material properties [31,32,33,34,35].

Additionally, orthopedic implants are subject to mechanical and chemical corrosion once implanted. This process releases metal ions from the implants which can form complexes with host cells surrounding the adjacent joint tissue. These protein–ion complexes can combine with host tissue to form antigen-presenting complexes that activate T helper cells of the immune system. Once activated, these cells coordinate the release of numerous inflammatory cytokines such as interleukins, tumor necrosis factors alpha, interferons, prostaglandins, and receptor activators of nuclear factors. These cytokines act as chemoattractants of immune cells and osteoclasts around the implants, leading to localized osteolysis, loosening, and pain for the patient. This phenomenon, known as metallosis, underscores the importance of selecting corrosion-resistant materials and adhering to stringent biocompatibility standards to minimize adverse reactions and enhance the longevity of the implant [47].

### 3.6. Cleanliness

Maintaining cleanliness during the manufacturing and handling of knee joint prostheses is critical to prevent postoperative infections, which can be catastrophic for the patient. This is especially pertinent for custom implants, which may involve more complex manufacturing processes and handling steps.

The ISO 13485 certification outlines the requirements for a comprehensive quality management system for the production of medical devices, including protocols for ensuring cleanliness and sterility. Stringent cleanliness standards are even more critical for implants with porous structures that enhance osseointegration. These structures are prone to collecting debris and microorganisms, which can lead to infections. Certified cleanliness protocols must include rigorous cleaning and sterilization processes to eliminate contaminants and reduce infection risks [41,47].

### 3.7. Toward Veterinary-Specific Implant Regulations: Challenges and Future Directions

Unlike human medicine, where implants are strictly regulated by ISO and ASTM standards, veterinary implants remain largely unregulated. Currently, no dedicated legal framework exists to guide the manufacturing, testing, and clinical evaluation of veterinary orthopedic implants, leading to inconsistencies in quality and safety. Reliance on human implant standards has been beneficial in providing guidelines for material selection, mechanical performance, and biocompatibility. Still, these standards do not fully address species-specific anatomical and biomechanical factors. As veterinary orthopedics advances and the demand for standardized implants grows, the need for formalized veterinary-specific regulations becomes increasingly evident.

One of the primary challenges in establishing such regulations is the vast variability in patient sizes and anatomical structures dealt with in veterinary medicine. While human implants are developed within a relatively narrow size range, veterinary implants must accommodate patients ranging from 1.5 kg toy breeds to 75+ kg giant breeds—a staggering 50-fold difference. This variability complicates standardized mechanical testing, load-bearing requirements, and the fit of implants, making applying a uniform regulatory framework across all veterinary species difficult.

Another critical issue is the lack of large-scale clinical trials and post-market surveillance in veterinary implantology. Unlike human medicine, where the long-term performance of implants is rigorously monitored, veterinary implants are often introduced based on limited case studies and small-scale research. The absence of mandatory reporting systems for complications, failure rates, and patient outcomes hinders the ability of manufacturers to refine implant designs based on real-world data. Introducing centralized implant registries for veterinary orthopedic implants—similar to those used in human joint replacement research—would provide valuable insights into long-term success rates and areas for improvement.

To address these challenges, the development of veterinary-specific implant standards should focus on three key areas:Material and biomechanical standards: Establishing species-specific guidelines for material properties, mechanical strength, surface roughness, and biocompatibility. Adaptations of ISO 5832 (metallic materials), ISO 10993 (biological evaluation), and ISO 7207 (knee prosthesis components) can serve as a foundation but must be refined to suit veterinary needs [31,32,33,34,35,36,37,38,39,40];Size-specific testing and classification: Given the extreme range of patient sizes involved in veterinary medicine, regulatory bodies should introduce size-based mechanical testing categories, ensuring that implants perform reliably across different weight classes. Load-bearing requirements should reflect the quadrupedal gait of animals, which differs from human bipedal locomotion;Quality control and post-market surveillance: A standardized certification process for veterinary implant manufacturers should be developed, requiring pre-clinical testing, long-term monitoring, and complication reporting. A collaborative effort between regulatory agencies, veterinary orthopedic societies, and manufacturers would be necessary to implement these measures effectively.

Establishing veterinary-specific implant regulations will require time and interdisciplinary collaboration. Still, it is a crucial step toward ensuring higher quality, safety, and predictability in veterinary orthopedic implants. In the meantime, the continued adaptation of human implant standards, and evidence-based refinements, offers the best available framework for advancing veterinary implantology.

## 4. Materials for Implants

Patellar groove replacement implants, in veterinary medicine, are used for restoring the joint function and alleviating discomfort in animals suffering from severe patellar luxation or degenerative knee conditions. The materials used in these implants must provide optimal biomechanical properties, exceptional durability, and biocompatibility to ensure long-term success and minimize complications.

In human medicine, there is no direct equivalent to patellar groove replacement implants; such implants are not used for human patients. However, the closest analogous implant in terms of its biomechanics, materials, and design is the human knee implant, particularly the femoral component. Human knee implants are designed to withstand similar joint forces and stresses, making them a relevant benchmark for veterinary patellar groove replacements.

Given the legal context, in which veterinary implants are not classified as medical devices, no specific regulatory standards govern their design and material properties. As a result, this study relies on well-established standards for human knee implants to guide its material and technical analysis. Using these standards as a reference, veterinary implants can be evaluated and developed with the same precision and reliability as in human orthopedic practice.

### 4.1. Metallic Alloys

Metallic alloys such as Co29Cr6Mo and Ti6Al4V are frequently used in veterinary patellar groove implants. Co29Cr6Mo alloy is valued for its high wear resistance and durability, which make it ideal for the demanding mechanical environment of the knee joint. It incorporates Cr_23_C_6_ carbides, which contribute to its robustness [48]. Ti6Al4V alloy is favored for its excellent biocompatibility and mechanical properties, which make it suitable for custom implant manufacturing via additive processes that allow for tailored solutions to specific clinical conditions [48] (Figure 4).

### 4.2. Ceramic Materials

Ceramic materials, notably zirconia, have been explored for their superior biocompatibility and wear resistance, which are highly beneficial traits in the context of total knee replacements in human medicine, and they are gaining interest regarding their use in veterinary applications. Ceramics offer reduced wear debris and a lower risk of allergic reactions, which is advantageous for long-term implant success [49]. Despite these benefits, ceramics’ brittleness and susceptibility to fracture under high-impact loads pose challenges. Veterinary implant designs must ensure adequate component thickness and stability, particularly in active animals which exert greater dynamic forces on the joint [50] (Figure 4).

### 4.3. Polymeric Materials

Polyetheretherketone (PEEK) and its composites are emerging as valuable materials for veterinary patellar groove replacements. PEEK’s mechanical properties closely mimic those of cortical bone, making it particularly suitable for withstanding the dynamic and variable loads experienced in the joints of animals. Studies indicate that PEEK can perform comparably to traditional metallic materials like CoCr regarding its mechanical integrity and fixation strength but offers a more favorable stress distribution in the surrounding bone tissue, potentially reducing stress shielding and preserving the bone quality [28,51]. However, PEEK may undergo more significant deformations under load than metallic components, which could influence its long-term durability and bone integration [52]. PEEK-OPTIMA™, a specific formulation of PEEK, has shown promising wear performance against polyethylene components, suggesting its viability as a load-bearing surface material in veterinary implants [53] (Figure 4).

### 4.4. Biomechanical Considerations

The choice of material significantly affects patellar implants’ biomechanical integration and performance. For example, titanium alloys can induce higher bone stresses, which might reduce the stress shielding compared to CoCr alloys. This aspect is crucial for encouraging natural bone remodeling around the implant [54]. Conversely, the excellent wear properties and lower debris generation of ceramic materials minimize the risk of inflammatory responses and enhance the longevity of the implant (Table 2).

### 4.5. Material Performance of Custom vs. Off-the-Shelf Implants

The selection of implant materials plays a critical role in the long-term success of orthopedic reconstructions. Various studies have compared titanium, cobalt-chromium, and bioceramic materials in custom implant applications. While titanium alloys remain the most commonly used due to their superior biocompatibility, corrosion resistance, and mechanical properties, cobalt-chromium alloys offer enhanced wear resistance, making them a viable alternative for high-load applications [55,56].

Advanced polymer–ceramic scaffolds, such as polycaprolactone/β-tricalcium phosphate composites, have been introduced as a potential alternative to conventional bone grafts, showing promising results in promoting osteointegration and structural stability. Additionally, synthetic osteoconductive materials and bone graft substitutes have been explored as complementary solutions in custom implant design [57,58,59].

Large-scale comparative studies on custom-designed vs. off-the-shelf implants in veterinary orthopedics remain scarce. However, case reports and smaller clinical studies suggest that customized implants provide superior anatomical compatibility and reduce the number complications associated with implant misalignment and instability. The use of custom acetabular prostheses in total hip replacements has shown improved clinical outcomes compared to traditional implant designs [4,60,61].

As additive manufacturing continues to evolve, further studies are needed to establish standardized guidelines for integrating 3D-printed custom implants into veterinary practice. Future research should focus on clinical trials that provide cost-benefit analysis and evaluate the long-term functional outcomes and complication rates of patient-specific solutions compared to conventional implants [62].

### 4.6. Implants Materials Summary

In veterinary patellar groove replacement, the selection of the implant materials—from Co29Cr6Mo and Ti6Al4V alloys to newer ceramic and polymeric options like PEEK—plays a pivotal role in the success of the surgery. Each material offers distinct advantages, such as durability, biocompatibility, and reduced wear, which are essential for the implant’s longevity and functional integration with the animal’s joint. However, the inherent challenges of ceramics and polymers, including their brittleness and potential for more significant deformations, highlight the necessity of ongoing development and clinical evaluation.

## 5. Surface Modifications—Articulating Joint Surface

Surface modifications of the articulating joint enhance knee implants’ performance and longevity. This section will explore the application and benefits of diamond-like carbon (DLC) and titanium nitride (TiN) coatings, which improve wear resistance, reduce friction, and increase biocompatibility.

### 5.1. Diamond-like Carbon

DLC coatings have been extensively studied for their potential to improve joint surface implants’ biocompatibility and wear resistance. DLC coatings are characterized by their hardness, low friction coefficient, and chemical inertness, which are beneficial for orthopedic applications [63,64,65]. Additionally, incorporating elements such as zirconium (Zr) into DLC coatings has enhanced the load-carrying capacity, adhesion, and tribological performance of implants, reducing the wear and friction in simulated body fluids [66]. However, there are challenges associated with the use of DLC coatings. High internal stress can lead to the poor adhesion of thick coatings, for which adhesion is a critical factor for load-bearing applications [67]. Depending on the deposition conditions, variation in the structure and properties of DLC coatings can affect their performance, and DLC tends to crack and delaminate under heavy load conditions [68]. Moreover, contradictory results have been reported regarding the effectiveness of DLC coatings in reducing wear in joint simulators, necessitating further research to clarify their use [64,69].

In summary, while DLC coatings offer promising attributes for use in joint surface implants, such as improved wear resistance and biocompatibility, the inconsistency of results and the technical challenges related to their adhesion and durability under load-bearing conditions require further investigation. Future research should focus on optimizing the existing deposition techniques and exploring the effects of doping elements to enhance the mechanical and biological performance of DLC coatings for joint implants [66,67,68,69] (Figure 5).

### 5.2. Titanium Nitride

TiN coatings are increasingly recognized for their potential to enhance the surface properties of joint implants. The application of TiN coatings to titanium and its alloys, such as Ti6Al-4V, has been shown to improve the wear resistance and reduce the ion release of the material, critical factors for joint replacements’ longevity and biocompatibility [70,71]. Notably, the powder immersion reaction assisted coating (PIRAC) method has been effective in producing adherent TiN coatings that exhibit lower residual stresses then physical vapor deposition (PVD) layers and demonstrate a significant reduction in their fretting wear and corrosion potential in vitro [72]. However, there are contrasting findings regarding the efficacy of TiN coatings. While some studies report enhanced wear resistance and reduced metal ion release [70,71], others suggest that the substrate material’s hardness may influence the TiN layers’ performance, indicating that further research is needed to optimize coating processes [73].

In summary, TiN coatings on joint implants show promise for improving wear resistance and reducing ion release, which are beneficial for the longevity of implants and patient outcomes. However, the performance of these coatings may be influenced by the underlying substrate material. The current literature underscores the need for continued research to optimize TiN coating techniques and fully understand their long-term clinical implications [70,71,72,73,74] (Figure 5).

## 6. Design and Surface Modifications for Osteointegration and Bone Ingrowth

The importance of the porosity in joint implants for osteointegration is highlighted by its role in facilitating bone growth and enhancing the mechanical and chemical connection between the implant and bone. Porous materials, as indicated in the research, allow the bone to grow around the implant’s pores, which can lead to an increase in the mass of the bone and the number of connection points, thereby improving the integration of the implant with the bone [75,76].

### 6.1. Pore Size

The optimal pore size of orthopedic implants for bone ingrowth has yet to be definitively established, as various studies have reported successful bone ingrowth with different pore sizes. For instance, some research suggests that a pore size of 600 μm in porous titanium implants manufactured by selective laser melting (SLM) is suitable for orthopedic applications due to the appropriate mechanical strength, high fixation ability, and rapid bone ingrowth of the resulting implants [77]. Another study indicates that a pore size of 178 μm in porous Ti6Al4V implants fabricated by coagulant-assisted foaming significantly promotes osteogenesis [78].

Further, successful bone formation has been reported in porous titanium implants with pore sizes of 800 and 1200 μm [79]. Additionally, a bionic trabecular structure with a pore size distribution of 200–1200 μm and an average pore size of 700 μm is suggested to be optimal for critical bone defect repair [80].

Moreover, more significant bone ingrowth in 3D-printed acetabular implants has been reported without an optimal pore size being specified [81]. Contradictions arise as different studies report varying optimal pore sizes, ranging from 178 μm to 1200 μm, for bone ingrowth. This discrepancy may be attributed to varying materials, manufacturing techniques, and biological environments across studies. The importance of a bionic trabecular structure, which includes a range of pore sizes rather than a single optimal size, is emphasized across the analyzed studies [80].

In summary, while there is no consensus on a single optimal pore size for bone ingrowth in orthopedic implants, a range between 178 μm and 1200 μm appears to be effective across different studies. The variability in the optimal pore size may be influenced by factors such as the implant material, the manufacturing technique, and the specific application within orthopedic surgery. Further research is needed to establish a standardized optimal pore size that maximizes bone ingrowth while maintaining mechanical integrity and promoting osseointegration [77,78,79,80,81,82] (Figure 6).

### 6.2. Porous Coatings

Porous coatings have garnered significant interest due to their ability to facilitate osteointegration. Porous coatings facilitate tissue ingrowth, which is essential for stable implant fixation [83]. Addressing challenges associated with porous coatings, such as ensuring a uniform thickness and strong adhesion to the implant surface, is critical to achieving long-term success [84]. Additionally, while porous metal surfaces have evolved to encourage bone apposition, there is a need for these implants to integrate biological and chemical methods, such as inducing stem cell growth and osteogenic differentiation, to further improve outcomes [85].

#### Types of Porous Coatings

1.Plasma-sprayed coatings

(a)Titanium and titanium alloy coatings—plasma spraying is used to apply a layer of titanium or titanium alloy onto the surface of implants. These coatings provide a rough, porous surface, promoting bone cell attachment and growth (Figure 7);(b)Hydroxyapatite (HA) coatings—hydroxyapatite, a naturally occurring mineral form of calcium apatite, is plasma-sprayed onto implants to create a bioactive surface that encourages direct bone bonding (Figure 7);

2.Metal bead coatings:

Tiny beads of metal, typically titanium or cobalt-chrome, are sintered (fused) onto the implant surface to create a porous structure. This enhances mechanical interlocking with the bone (Figure 7);

3.Metallic fiber mesh coatings:

A mesh of fine metallic fibers, usually titanium or cobalt-chrome, is sintered onto the implant. This mesh provides a large surface area for bone ingrowth and mechanical solid interlocking (Figure 7);

4.Porous tantalum trabecular metal:

Tantalum is used to create a highly porous and biocompatible surface. Porous tantalum coatings have excellent osteoconductive properties, promoting rapid bone ingrowth and strong fixation (Figure 7);

5.Bioactive glass coatings:

Bioactive glass materials are applied as a coating to create a porous, bioactive surface that supports bone bonding and healing. Bioglass coatings can be tailored to provide varying degrees of porosity and bioactivity;

6.Tricalcium phosphate (TCP) coatings:

Like hydroxyapatite, tricalcium phosphate is a porous coating that is used to enhance bone ingrowth. TCP coatings are resorbable, meaning they gradually dissolve and are replaced by natural bone (Figure 7).

**Figure 7 materials-18-01652-f007:**
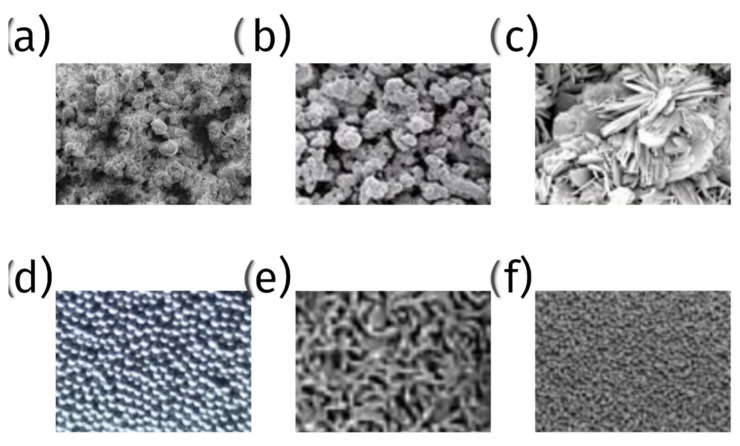
Microscopic pictures of surface coatings—(**a**) hydroxyapatite plasma spraying, (**b**) titanium alloy plasma spraying, (**c**) tricalcium phosphate, (**d**) cobalt-chrome alloy beads, (**e**) titanium fiber mesh, (**f**) porous tantalum.

### 6.3. 3D Printed Porous Structures

Advanced manufacturing techniques, such as additive manufacturing, allow for the creation of tailored porous implants that can reduce stress shielding and promote better bone ingrowth, offering unparalleled precision and customization. This technology enables the creation of sophisticated topographies that are essential to achieving optimal osseointegration and mechanical stability, which traditional manufacturing techniques cannot achieve [86,87,88].

#### 6.3.1. Techniques and Materials

Selective laser melting (SLM) and electron beam melting (EBM) are the principal 3D printing methods for fabricating metal porous structures for orthopedic applications (Table 3).

#### 6.3.2. Advantages and Challenges

3D printing, in fabricating metal porous structures for orthopedic implants, offers several advantages. The precise control that this method offers over the pore size and distribution of lattices made from Co-Cr-Mo and titanium alloys facilitates bone growth within their structure, leading to improved implant stability and longevity [76,88]. Interestingly, the design of porous structures can vary, with gradient porous designs mimicking natural bone morphology more closely than uniform porous structures. These gradient designs are considered more promising for orthopedic applications due to their improved mechanical properties [88]. The customization of these designs allows producing implants that precisely fit the patient’s unique anatomy, thereby enhancing surgical outcomes. Additionally, the reduced lead time associated with 3D printing accelerates the transition from design to production, enabling the rapid delivery of patient-specific solutions (Figure 8).

However, several challenges accompany these advantages. Ensuring consistent quality and mechanical properties across different batches presents a significant hurdle in the quality control of implants made using this method. Navigating the evolving regulatory landscape for 3D-printed medical devices poses another challenge to obtaining regulatory approval. Furthermore, the high costs associated with the equipment and materials needed for 3D printing may limit its widespread adoption in the field.

## 7. Current Market Review

The domain of patellar groove replacement (PGR) has experienced substantial advancements in recent years, propelled by innovations in materials science, manufacturing technologies, and surgical methodologies. This summary comprehensively analyzes the extant PGR products and technologies market, providing a detailed overview of leading manufacturers and their respective offerings. It further delves into emerging trends and innovations poised to influence the future trajectory of patellar groove replacements. Moreover, this chapter identifies and examines the prevailing challenges and opportunities within the market, offering critical insights into the factors that impact the adoption and evolution of these essential veterinary orthopedic solutions.

### 7.1. KYON Patellar Groove Replacement (PGR)

The KYON patellar groove replacement (PGR) system is a two-component solution designed to address patellar luxation, a condition characterized by the dislocation of the patella from the patellar groove.

The system consists of two main components:Groove component: This element is fabricated from a titanium alloy and coated with amorphous diamond-like carbon (ADLC). The ADLC coating imparts a smooth, complex, and scratch-resistant surface with a low coefficient of friction, thereby minimizing the heat generated by interaction with the surface;Base plate: constructed from commercially pure titanium and coated with calcium phosphate, this component is designed to facilitate integration with the bone.

The KYON PGR system presents several advantages over traditional surgical techniques for addressing patellar luxation:-Immediate stability: the implant provides immediate stability for the patella within the artificial groove, reducing the risk of luxation;-Low friction: the ADLC coating on the groove component ensures smooth patellar movement, minimizing friction and wear;-Rapid bone ingrowth: the BioCer^®^ surface treatment on the base plate promotes rapid bone ingrowth, enhancing implant stability;-Size options: the KYON PGR is available in over 10 sizes, allowing surgeons to select the most appropriate fit for various dog breeds.

### 7.2. Innoplant TRA

The Innoplant TRA (trochlear ridge arthroplasty) represents a one-piece implant that is designed to treat patellar luxation in dogs, accommodating a wide range of sizes, including small breeds as light as 1.5 kg.

The Innoplant TRA. implant features a one-piece design with a plug for force distribution. It is fabricated from a titanium alloy (TiAl6V4) that complies with the ISO 5832-3 and ASTM F1472 standards [89], ensuring high-quality material performance.

The implant’s surface is characterized by an open-pored, lattice-shaped design that promotes osteointegration, facilitating bone growth into the implant. The contact surface for the patella is polished and coated with diamond-like carbon (DLC) to ensure smooth articulation and reduce friction.

Available in a comprehensive range of sizes from #1 to #10, including intermediate sizes, the Innoplant TRA can accommodate various patient anatomies. Additionally, the three small sizes (from #1 to #3) are available in a broader version to address a wider spectrum of clinical indications.

### 7.3. Surgical Procedures

The surgical procedure for the KYON PGR involves preparing a bone bed for the base plate, and entails removing a section of bone from the patellar groove to create a broad, well-vascularized surface. The base plate is then affixed to this area using titanium bone screws. Subsequently, the groove component is secured to the base plate via conical pegs. This system offers considerable flexibility in positioning the base plate, allowing surgeons to optimize the alignment of the quadriceps tendon and patella. This adjustability can prevent the need for tibial tuberosity transposition, a procedure frequently performed to correct patellar luxation.

The primary anchorage of the Innoplant TRA is achieved through a combination of press-fit, form-fit, and screw-fixation techniques, ensuring secure placement within the bone. Secondary anchorage is facilitated by osteointegration, wherein new bone stock forms around the implant, enhancing its long-term stability and fixation.

### 7.4. Current Off-Shelf Products’ Disadvantages and Limitations

The KYON PGR system, while offering numerous advantages, has certain limitations. One significant drawback is the necessity for manual trochlear osteotomy cutting without the aid of specific surgical guides. This reliance on the surgeon’s experience and skill can limit the accuracy of the procedure, potentially affecting the alignment and stability of the implant. Additionally, the complexity of the procedure can be heightened in small patients, where precision is even more critical.

Similarly, the Innoplant TRA implant faces challenges related to the manual cutting required during the surgical procedure. The absence of specialized guides means that the accuracy of the bone preparation and implant placement depends on the surgeon’s expertise. This can lead to variability in surgical outcomes. It may also complicate the procedure, particularly in small patients for whom there is a minimal margin for error and the anatomical structures are more challenging to navigate.

### 7.5. Challenges in Maintaining Anatomical Surface Levels

One critical challenge with off-the-shelf implants is achieving the correct surface level of the patellar groove component. In natural anatomy, the depth and contour of the trochlear groove ensure proper patellar tracking and articulation. However, the fixed dimensions of standardized implants often result in an elevated groove bottom relative to the original anatomy (Table 4). This misalignment can lead to altered joint biomechanics and increased wear on the patella.

### 7.6. Implications for Patella Wear

In cases where the implant thickness exceeds the natural groove depth, the patella is subjected to abnormal loading patterns, accelerating wear and potentially leading to secondary complications. Conversely, an insufficient implant thickness can compromise the stability of the patella within the groove, increasing the risk of luxation or subluxation. Additionally, surgical limitations can further complicate achieving the desired groove depth. In some cases, cutting the distal femur deeply enough to replicate the natural groove-bottom level is not feasible, as it may compromise the structural integrity of the femur. An insufficient amount of remaining bone can weaken the distal femur, increasing the risk of fracture or implant failure. Achieving an implant groove-bottom surface level that mirrors the original anatomical dimensions is particularly challenging with current off-the-shelf options, underscoring a significant limitation in their design [90,91].

### 7.7. Implant Placement Challenges and Biomechanical Considerations

Proper alignment of the patellar groove replacement implant is critical for maintaining the biomechanical function of the joint. The groove axis must align precisely with the natural tracking of the patella and the direction of the forces exerted by the quadriceps tendon. Misalignment, even by a small degree, can disrupt the smooth articulation of the patella within the groove, leading to abnormal wear patterns and increased stress on the surrounding soft tissues.

The use of available off-the-shelf implants relies heavily on the surgeon’s experience and intraoperative judgment to achieve proper alignment. During surgery, the surgeon assesses the joint function and implant positioning subjectively, often through manual manipulation and visual inspection. This method introduces variability, as even experienced surgeons may find it challenging to ensure perfect alignment, especially in complex cases where anatomical landmarks are distorted or absent due to the pathology or previous interventions.

Misaligned implants can result in an uneven distribution of forces across the patella, accelerating wear and potentially leading to complications such as patellar luxation, chronic pain, or a reduced range of joint motion. Furthermore, deviations in the groove’s axis can alter the quadriceps mechanism’s load dynamics, exacerbating mechanical inefficiencies and increasing the likelihood of implant failure.

To address these challenges, advancements in preoperative planning tools, such as 3D imaging and modeling, and the development of intraoperative guidance systems could provide surgeons with more objective methods for assessing joint function and ensuring proper implant placement. Such innovations would reduce the reliance on subjective assessment, improving the consistency of surgical outcomes and minimizing the biomechanical risks associated with patellar groove replacement implants.

## 8. Advances in Technology

The field of patellar groove replacement has seen significant technological advancements, particularly with the advent of custom-made implants that can be manufactured using metal 3D printing technologies. These advancements offer a range of benefits, especially in cases where standard implants may be inadequate due to anatomical complexities or severe pathological conditions.

### 8.1. Advantages and Potential of Custom Patellar Groove Implants

Custom patellar groove implants (cPGIs) provide distinct advantages by addressing the unique anatomical needs of individual patients, a capability that is particularly beneficial in veterinary orthopedics. 3D printing enables the creation of implants that are precisely tailored to the patient’s specific anatomy, which is invaluable in cases involving bone loss, deformities, or complex revisions. This level of customization enhances the fit and function of the implant, improving surgical outcomes and patient recovery [92] (Figure 9).

One of the critical benefits of cPGIs is the ability to incorporate porous structures within the implant. These porous designs promote osseointegration, where bone tissue grows into the implant, ensuring long-term stability and significantly reducing the risk of loosening compared to that of standard implants. This integration is crucial to the durability and functionality of the implant, particularly in active patients [18] (Figure 10).

Custom implants are especially promising in the treatment of challenging conditions such as osteomyelitis, which can result in significant bone defects. Custom solutions restore joint function and provide a stable scaffold for bone regeneration by designing implants that precisely fill these gaps. This approach can prevent the need for more drastic measures like amputation, thereby offering patients a significantly improved quality of life [18].

Another critical advantage of cPGIs is the ability to design them for patient-specific screw placement. In situations where anatomical landmarks are altered or compromised, this customization ensures secure fixation, reducing the risk of complications such as implant migration or failure. Standard pre-made implants cannot offer this level of precision, often leading to suboptimal outcomes [18] (Figure 11).

The capability to create complex implants for parallel corrective osteotomy is another significant advantage of 3D printing—corrective custom patellar groove implants (ccPGIs). These allow for the simultaneous correction of multiple anatomical deformities, further enhancing the functional outcomes of the surgery [92].

Moreover, cPGIs can be tailored to consider a broader range of size parameters than off-the-shelf products. Factors such as the patellar width, length, thickness, groove shape, and depth can all be precisely tailored. This comprehensive customization ensures that the implant fits perfectly within the patient’s anatomical structure, enhancing the immediate and long-term success of the surgery.

### 8.2. Technological Implementation and Future Directions

In custom patellar groove implants, 3D printing plays a crucial role in the fabrication of the implant and the production of comprehensive surgical sets. These sets include custom surgical guides and patient-specific bone models, which collectively enhance the precision and efficacy of surgical procedures.

The custom surgical guides are designed to provide the highest level of accuracy during patellar groove replacement surgical procedures, such as trochlear osteotomy or trochlear osteotomy with parallel corrective distal femur osteotomy. Tailored to the patient’s unique anatomy, these guides significantly enhance surgical precision, improving outcomes. This precision mainly benefits small patients, where anatomical nuances require meticulous attention (Figure 12).

Additionally, the surgical set will include patient-specific bone models created through 3D printing. These models serve as vital tools for surgical training, allowing surgeons to practice and meticulously plan the surgery before the actual procedure. This preparatory step reduces the likelihood of errors, enhances surgical efficiency, and improves overall surgical effectiveness (Figure 13).

The surgical sets used for custom patellar groove implants ensure a comprehensive approach to veterinary orthopedic surgery by incorporating custom surgical guides and training bone models. This integration of advanced 3D printing technologies enhances the fit and function of implants. It supports the entire surgical process, from planning to execution, thereby improving patient outcomes and advancing the field of veterinary medicine.

### 8.3. Challenges and Considerations

Despite the numerous advantages, several challenges are associated with using 3D-printed cPGIs. One significant issue is process consistency. Since each implant must be uniquely designed and manufactured, maintaining the highest quality in terms of polishing and mechanical strength can be more complicated than with mass-produced implants. Therefore, selecting an implant provider with proven consistency in these aspects and robust process certification is crucial (Figure 14).

Furthermore, the design of these custom implants requires highly skilled engineers. Each patient presents a unique set of anatomical features and requirements, and in veterinary medicine, the size difference between the smallest and largest patients can be more than 50-fold. This variability demands a high level of expertise in designing prostheses that meet each patient’s needs.

The development and use of custom surgical guides are equally crucial in ensuring the precision and success of surgical procedures. These guides facilitate accurate drilling and cutting, which are essential to the proper placement of the implants. However, the material used to manufacture these surgical guides must be meticulously chosen. The selected material must exhibit excellent biocompatibility to prevent adverse reactions during and after surgery and possess the mechanical properties necessary to withstand the forces involved in precise drilling and cutting. Materials that meet these stringent requirements could improve the surgery’s accuracy and the patient’s overall outcome.

#### Risks of Combined Prosthesis/Plate Implants

Combined prosthesis/plate implants (corrective custom patellar groove implants, ccPGIs), while often necessary for complex reconstructive procedures, introduces specific risks. One significant disadvantage is the interdependence of the prosthesis and the plate. In cases where the plate must be removed due to complications such as infection or screw loosening, maintaining the prosthesis becomes challenging. This scenario may necessitate the removal of both components, even when the prosthesis is functioning well. The need for complete removal can result in additional surgical interventions, prolonged recovery periods, and increased costs.

Moreover, the presence of a combined implant system may increase the complexity of the surgical procedure, especially in the hands of inexperienced surgeons. The accurate placement of both components requires meticulous planning and execution to avoid biomechanical mismatches or unintended strain on the implant. These risks emphasize the need for advanced preoperative planning tools and intraoperative precision, such as that provided by custom surgical guides, to mitigate complications.

In conclusion, developing custom patellar groove implants through metal 3D printing technologies significantly advances veterinary orthopedics. These implants offer tailored solutions that enhance their fit, function, and integration, allowing veterinarians to address the unique needs of individual patients and improving overall surgical outcomes. The ability to consider a wide range of size parameters, create complex implants for parallel corrective osteotomy or reconstructive surgery, and use 3D-printed custom surgical guides and bone models for surgical training further highlights the transformative potential of this technology. However, challenges such as ensuring process consistency and the requirement for highly skilled engineers must be addressed to realize the benefits of custom-made implants. As technology continues to evolve, its role in treating complex orthopedic conditions will undoubtedly become more prominent, offering new possibilities for enhancing the quality of life of animal companions.

## 9. Conclusions

This review has underscored the importance of adopting human medical standards for the design, manufacturing, and quality assurance of veterinary implants, particularly patellar groove replacement implants. Given the absence of specific legal requirements and standards for veterinary implants, referencing human knee implant standards provides a robust framework to ensure safety, efficacy, and high-quality outcomes in veterinary applications.

The biomechanical and anatomical similarities between human and canine knee joints justify the application of human standards in veterinary medicine. Both species share common pathologies and similar joint structures, which influence the forces and pressures experienced by the knee joint. Studies have emphasized the role of dynamic forces and joint alignment in maintaining stability and function, which are equally relevant to canine joints. This supports the rationale for using human standards as a reference in veterinary applications.

The materials used for orthopedic implants, as specified by ISO standards, are critical to ensuring the biocompatibility and mechanical performance of the implant. The ISO 5832 series outlines the requirements for metallic materials used in surgical implants, ensuring that they offer the necessary mechanical properties and biocompatibility for the demanding environment of the knee joint. The ISO 10993 series further ensures comprehensive biological evaluations to confirm the safety of these materials for long-term implantation.

Surface roughness is a crucial factor in the performance of knee joint prostheses. According to ISO 7207-2, the articulating surfaces of metallic or ceramic femoral components must meet stringent surface roughness requirements to minimize friction and wear. However, the smaller size of veterinary implants necessitates adjustments to these standards to achieve the same level of precision and smoothness. Maintaining cleanliness during the manufacturing and handling of knee joint prostheses is critical to prevent postoperative infections, especially for custom implants with porous structures designed to enhance osseointegration.

Engineers can now manufacture custom-made implants via metal 3D printing technologies, offering notable advantages in complex cases where standard implants are unsuitable. These implants can be tailored to the unique anatomies of individual patients, offering an enhanced fit and enhanced function and integration. Additionally, porous structures within custom implants promote osseointegration, contributing to long-term stability and reducing the likelihood of loosening. However, while these benefits are significant, we must recognize that custom implants are not universally required.

For most cases, a series of standardized implants available in multiple sizes can address the clinical needs of the patient effectively and efficiently. Adopting a Pareto principle approach is often practical, as 80% of cases can be managed using standard-sized implants. In comparison, the remaining 20%—those presenting with the most complexity and accounting for 80% of the challenges—may benefit from custom solutions. This approach balances clinical effectiveness with economic viability, as custom implants are typically more expensive due to the specialized design, manufacturing processes, and postprocessing involved.

From an economic perspective, most veterinary surgeons cannot justify custom implants for routine cases when the standardized options provide equivalent outcomes. However, the quality of the implant, whether standardized or custom-made, must remain the highest priority. Superior material properties, precise design, and optimal surface roughness are critical to ensuring any implant’s safety, functionality, and longevity.

While advantageous in complex scenarios, custom solutions pose additional challenges in ensuring consistent quality, design precision, and surface roughness. Variability in the manufacturing process, particularly with 3D printing, of patellar implants necessitates rigorous quality control to meet the stringent requirements of human orthopedic implants. Therefore, surgeons should only select high-quality custom implants for clinical use because the use of a substandard design or finish can compromise surgical outcomes.

We recommend tailoring implant selection to each clinical scenario and reserving custom implants for cases in which they offer superior benefits. Regardless of the approach, ensuring the highest quality of materials, manufacturing processes, and surface roughness is essential to achieving optimal outcomes in veterinary orthopedics.

In conclusion, we urge veterinary implant developers to adopt human medical standards when creating patellar groove replacement implants, ensuring safer, more effective treatment options for animal patients. By leveraging these established standards, veterinary practitioners can improve surgical outcomes and enhance the quality of life of their patients. Future research should focus on quantifying and directly comparing the specific forces in human and canine knee joints to refine and validate the application of these standards in veterinary orthopedics.

## Figures and Tables

**Figure 1 materials-18-01652-f001:**
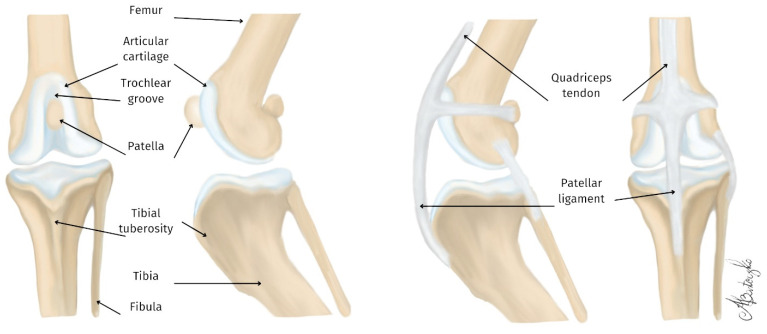
The canine stifle anatomy scheme.

**Figure 2 materials-18-01652-f002:**
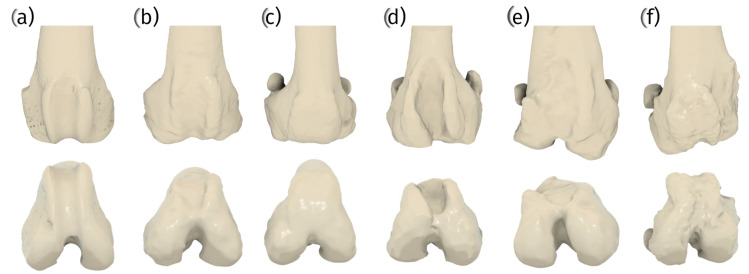
Examples of normal and pathological patellar grooves—(**a**) physiological, (**b**) asymmetrical, (**c**) flat, (**d**) after sulcoplasty, (**e**) unknown origin trauma, (**f**) severe osteoarthritis with parallel deformation.

**Figure 3 materials-18-01652-f003:**
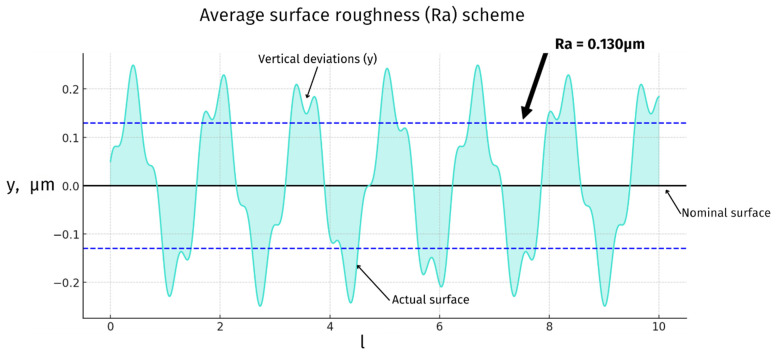
Surface roughness scheme (Ra—average surface roughness).

**Figure 4 materials-18-01652-f004:**
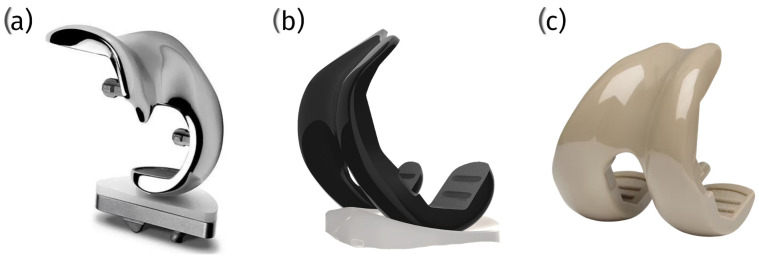
Representative materials, and their visual differences, for the femoral component of a human knee implant—femoral part—(**a**) metallic—cobalt-chrome alloy, (**b**) ceramic—Oxinium^®^, (**c**) polymeric—Peek Optima^®^.

**Figure 5 materials-18-01652-f005:**
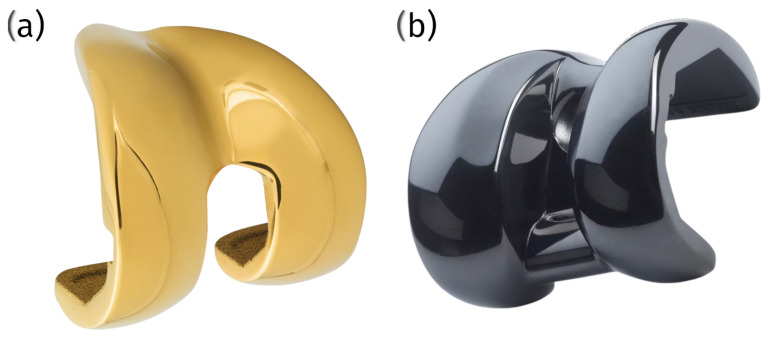
Schematic representation of surface coatings for enhanced wear resistance in human knee prosthesis—femoral part—(**a**) TiN coating, (**b**) DLC coating.

**Figure 6 materials-18-01652-f006:**
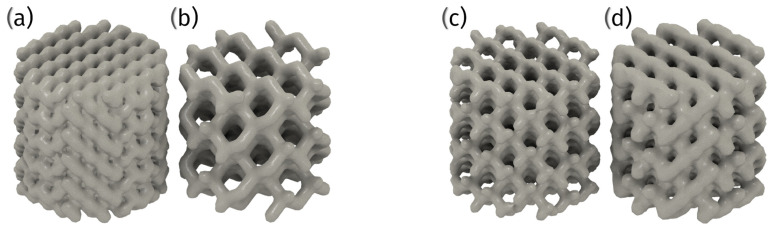
Schematic examples of porous lattice structures for orthopedic implants—(**a**,**b**) beam diameter the same, (**a**) pore size 0.5 mm, (**b**) pore size 2 mm. (**c**,**d**) Lattice shape the same, (**c**) beam size 0.8 mm, (**d**) beam size 1.2 mm.

**Figure 8 materials-18-01652-f008:**
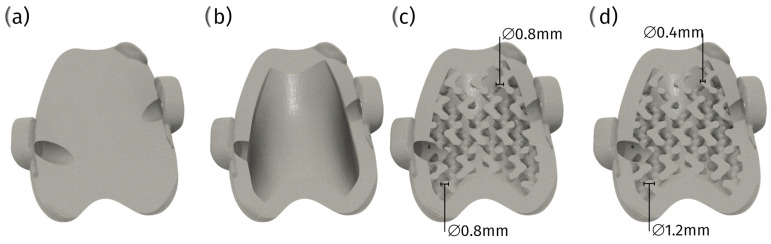
Different design approaches for 3D-printed patellar groove implants—(**a**) solid, (**b**) hollow, (**c**) same size porosity, (**d**) porosity size gradient.

**Figure 9 materials-18-01652-f009:**
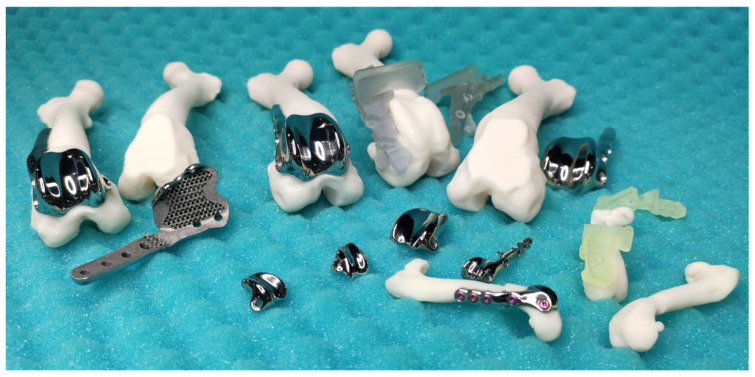
Patient-specific implants manufactured via 3D printing. Each cPGI design matches a dog’s unique anatomical and/or corrective requirements.

**Figure 10 materials-18-01652-f010:**
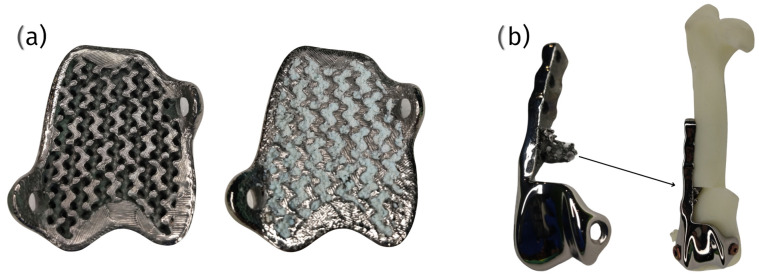
3D printed implants’ porosity—(**a**) simple prosthesis with porosity optimized for hydroxyapatite filling during surgery (empty porosity vs. HaP filled porosity), (**b**) complex corrective implant with porosity as a mechanical support for bone discontinuity.

**Figure 11 materials-18-01652-f011:**
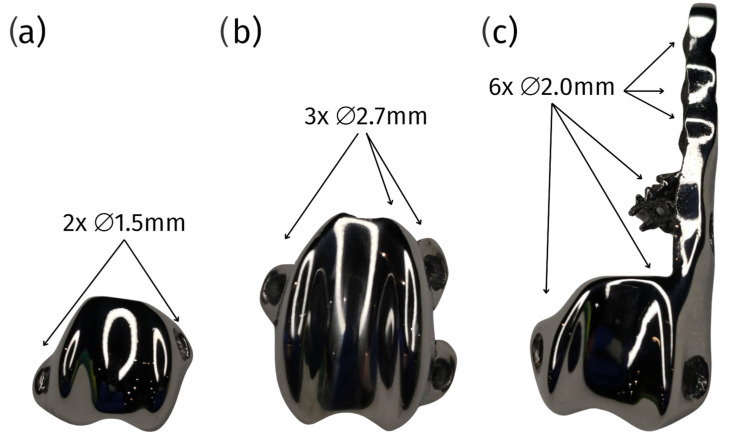
Patient-specific screw placement in 3D-printed patellar groove implants—(**a**) two-screw mounting, (**b**) three-screw mounting, (**c**) complex corrective implant with six-screw mounting.

**Figure 12 materials-18-01652-f012:**
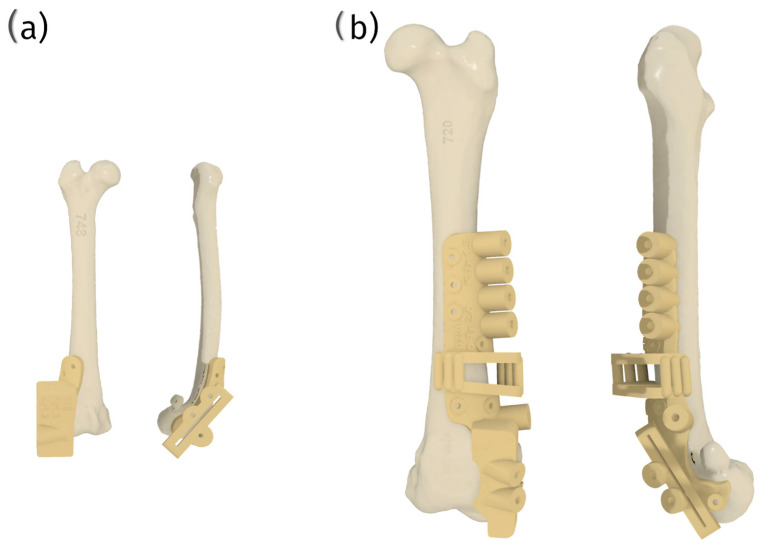
Surgical guides for precise patellar groove replacement—(**a**) simple cPGI cutting and drilling guide, (**b**) complex ccPGI cutting and drilling guide (torsion and aLDFA correction with parallel 5 mm femur shortening).

**Figure 13 materials-18-01652-f013:**
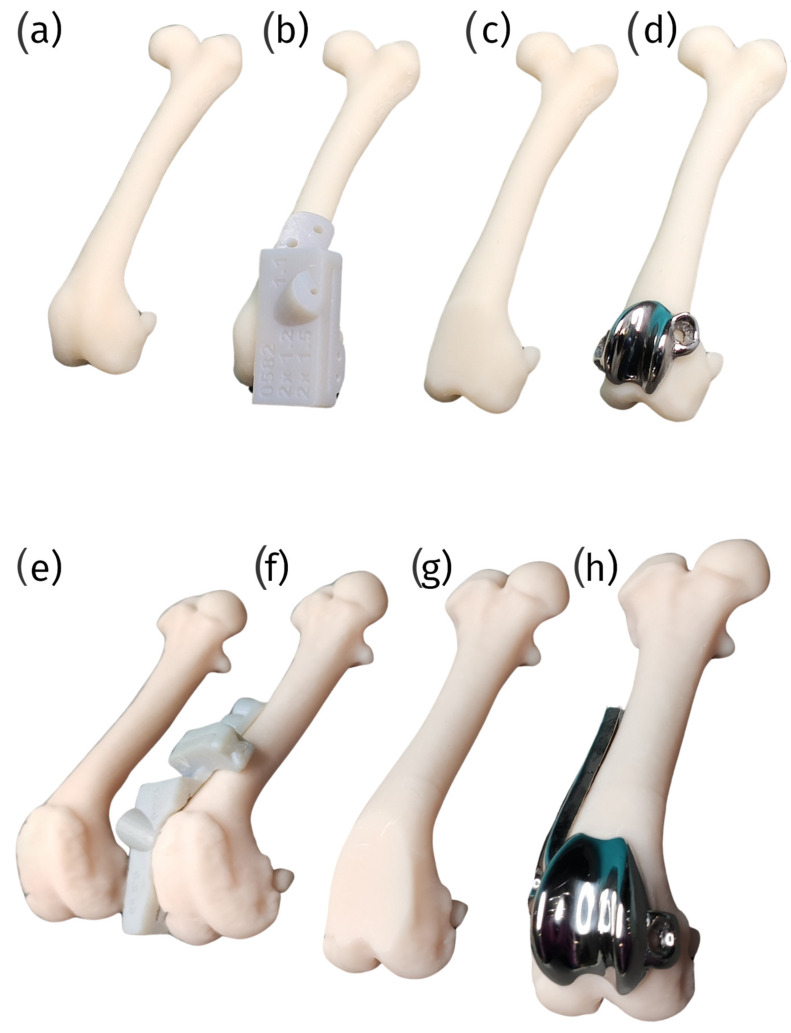
Complete 3D-printed surgical training and implementation set—(**a**–**d**) simple custom patellar groove implant, (**a**) training bone, (**b**) training surgical guide, (**c**) bone after cutting off the trochlea groove, (**d**) implant fitting trial. (**e**–**h**) Corrective custom patellar groove implant, (**e**) training bone, (**f**) training surgical guide, (**g**) bone after cutting off the trochlea and performing DFO (torsion change), (**h**) corrective implant trial.

**Figure 14 materials-18-01652-f014:**
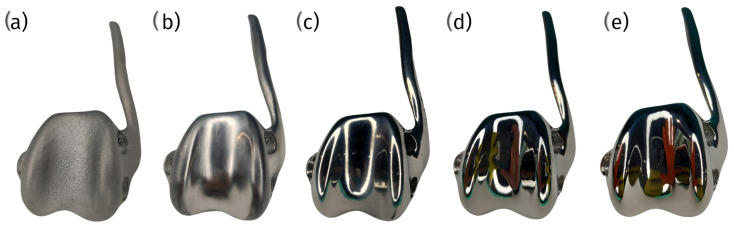
Titanium alloy, selective laser melting 3D printed implant postprocessing steps—(**a**) raw surface, (**b**) ground surface, (**c**) initially polished surface, (**d**) finishing polishing surface, (**e**) electrochemically polished surface with Ra_max_ = 0.023 µm (human medicine requirement Ra_max_ < 0.1 µm).

**Table 1 materials-18-01652-t001:** The chosen standards may apply to veterinary implants [31,32,33,34,35,36,37,38,39,40,41,42,43,44,45].

Standard Series	Standard
ISO 5832 This series specifies the requirements for metallic materials used to manufacture surgical implants.	ISO 5832-1: Implants for surgery—Metallic materials—Part 1: Wrought stainless steel.
	ISO 5832-2: Implants for surgery—Metallic materials—Part 2: Unalloyed titanium.
	ISO 5832-3: Implants for surgery—Metallic materials—Part 3: Wrought titanium 6-aluminium 4-vanadium alloy.
	ISO 5832-4: Implants for surgery—Metallic materials—Part 4: Cobalt-chromium-molybdenum casting alloy.
	ISO 5832-12: Implants for surgery—Metallic materials—Part 12: Wrought cobalt-chromium-molybdenum alloy.
ISO 1099 This series provides guidelines for the biological evaluation of medical devices, including orthopedic implants.	ISO 10993-1: Biological evaluation of medical devices—Part 1: Evaluation and testing within a risk management process.
	ISO 10993-5: Biological evaluation of medical devices—Part 5: Tests for in vitro cytotoxicity.
	ISO 10993-11: Biological evaluation of medical devices—Part 11: Tests for systemic toxicity.
ISO 7207 This series pertains to the components used in knee joint prostheses.	ISO 7207-1: Implants for surgery—Components for partial and total knee joint prostheses—Part 1: Classification, definitions, and designation of dimensions.
	ISO 7207-2: Implants for surgery—Components for partial and total knee joint prostheses—Part 2: Articulating surfaces made of metal, ceramic, and plastic materials. This part specifies surface finish requirements for the articulating surfaces of total and partial knee joint prostheses classified in ISO 7207-1.
ISO 13485: Medical devices—Quality management systems—Requirements for regulatory purposes	
ISO 14243 This series outlines the methods for testing the wear of knee joint prostheses.	ISO 14243-1: Implants for surgery—Wear of total knee-joint prostheses—Part 1: Loading and displacement parameters for wear-testing machines with load control and corresponding environmental conditions for the test.
	ISO 14243-2: Implants for surgery—Wear of total knee-joint prostheses—Part 2: Methods of measurement
ISO 21534: Implants for surgery—Joint replacement implants—Particular requirements.	
ISO 21535: Non-active surgical implants—Joint replacement implants—Specific requirements for hip-joint replacement implants. While specific to hip joint replacement, these principles and testing methods are also relevant to knee joint prostheses.	

**Table 2 materials-18-01652-t002:** Selected properties of materials used for knee implants—femoral part.

Material	Density	Biocompatibility	Wear Resistance	Bone Integration	Brittleness	Manufacturability/Price
Cobalt Alloys	~8.3 g/cm^3^	Good	Very High	Good	Low	Difficult/High
Titanium Alloys	~4.4 g/cm^3^	Good	Moderate	Very good	Low	Moderate/Moderate
Bioceramics (Zirconia)	~5.7–6.1 g/cm^3^	Very good	High	Very Good	High	Difficult/High
Polymers (PEEK)	~1.3 g/cm^3^	Good	Moderate	Poor	Low	Moderate/High

**Table 3 materials-18-01652-t003:** Metal 3D printing technologies comparison.

	Selective Laser Melting (SLM)	Electron Beam Melting (EBM)
Process	Utilizes a high-power laser to melt and fuse metal powder layer by layer selectively.	Employs an electron beam to melt metal powder in a vacuum environment, layer by layer.
Materials	Commonly includes titanium alloys (e.g., Ti-6Al-4V), cobalt-chromium alloys, and stainless steel.	Titanium alloys were chosen primarily for their biocompatibility and mechanical properties.
Advantages	Facilitates high precision and the creation of intricate lattice structures with controlled porosity, promoting enhanced mechanical properties and bone ingrowth.	Produces parts with excellent mechanical properties, suitable for high-performance implants with complex geometries.

**Table 4 materials-18-01652-t004:** KYON PGR measurements are based on the vPOP Pro software version 3.0.10 and implant template database (London, UK).

	Implant Measured Dimensions, mm			
Implant Size	Width	Length	Height (From Base to Groove Condyles Peak Surface)	Thickness (From Base to the Bottom of the Groove)
#1	6.5	11.5	5.5	3.7
#1.5	7.5	13.0	6.0	3.9
#2	8.5	14.5	6.5	4.1
#2.5	9.5	16.5	7.0	4.3
#3	10.5	18.0	7.5	4.6
#4	12.5	21.5	8.5	5.0
#5	14.5	25.0	9.5	5.5
#6	16.5	28.0	10.5	6.0
#7	18.5	32.0	11.5	6.5
#8	21.0	36.0	13.0	7.0
#9	23.5	40.5	14.5	7.5
#10	26.0	45.0	16.0	8.5

## Data Availability

Not applicable.

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
