# Peer review of "Evaluation of Patellar Groove Prostheses in Veterinary Medicine: Review of Technological Advances, Technical Aspects, and Quality Standards"

_materials, 2025, doi:10.3390/ma18071652_

Round 1
Reviewer 1 Report
Comments and Suggestions for Authors
In this review article, the authors of the paper discuss in detail the technical advancements, biomaterials engineering considerations, and quality standards of patellar groove protheses in veterinary medicine. The authors went in detail on how there is still a lack of veterinary-specific regulations when it comes to animal implants and how integrating human implant standards and benchmarks into consideration can help with veterinary implant applications in terms of material selection, biocompatibility, and mechanical performance. Overall, this review is well-researched, well-written, well-organized, and provides great insights into veterinary orthopedic implants. I only have a few minor comments for the authors:
- It would be useful to include a section reviewing clinical studies or case reports on patellar groove implants that have been applied in veterinary medicine.
- Are there any cases where patient-specific 3D printed orthopedic implants have been used in veterinary medicine? It would be helpful if real world case reports can be included in this paper, given that there are sections dedicated to 3D printing technology in this paper.
- Due to the biomechanical differences between human and animal knee joints, what are the extents of the applicability of human implant standards to those of animal?

Author Response
Dear reviewer, thank you for your valuable review. We have rewritten the paper according to your suggestions.
Comment 1: It would be useful to include a section reviewing clinical studies or case reports on patellar groove implants that have been applied in veterinary medicine.
Answer: We acknowledge the reviewer’s suggestion. However, the literature on veterinary patellar groove implants is extremely limited, with only two identified studies addressing off-the-shelf implants (Dokic et al., 2015) and another focusing on a custom-designed solution (Panichi et al., 2024). These studies are referenced and discussed in the manuscript to provide the most comprehensive overview.
Comment 2: Are there any cases where patient-specific 3D printed orthopedic implants have been used in veterinary medicine? It would be helpful if real world case reports can be included in this paper, given that there are sections dedicated to 3D printing technology in this paper.
Answer: We have added the subchapter "3D-Printed Patient-Specific Implants in Veterinary Medicine," which includes documented cases of custom implants used in patellar groove replacement, limb salvage, and joint reconstruction. This section highlights their clinical application and benefits in veterinary orthopedics.
Comment 3: Due to the biomechanical differences between human and animal knee joints, what are the extents of the applicability of human implant standards to those of animal?
Answer: The applicability of human implant standards to veterinary patellar groove replacements (PGRs) is feasible and highly beneficial. While biomechanical differences exist between human and animal knee joints, the fundamental principles governing implant durability, surface quality, and biocompatibility remain universal.
The human knee endures significantly greater forces due to bipedal locomotion, yet human implants have been designed to function under these demanding conditions for decades. In comparison, veterinary patients, particularly small animals, experience different but substantial joint stresses. If an implant meets human standards, it will unquestionably be sufficient for veterinary applications.
Surface quality is a key factor in implant performance, directly influencing cartilage wear. Regardless of species, poor surface finishing increases friction and accelerates cartilage degeneration, leading to pain and joint dysfunction. Adhering to human implant standards ensures optimal surface smoothness, reducing wear-related complications and improving long-term outcomes. While animals have shorter lifespans and do not require implants lasting 20+ years, they experience pain and discomfort just as humans do. High-quality implants should, therefore, be prioritized to restore function and ensure a good quality of life.
Rather than reinventing the wheel, veterinary orthopedics should leverage validated human implant standards to ensure safety, reliability, and efficiency. By adopting these well-established principles, we can improve surgical outcomes, minimize implant failure rates, and enhance the well-being of veterinary patients without unnecessary compromises.
Reviewer 2 Report
Comments and Suggestions for Authors
I consider that the subject of the paper deserves attention. The authors approach some technological aspects regarding prostheses applied in veterinary medicine. At this stage, the paper is more like a theoretical paper and perhaps it would be better to highlight more the results obtained by other authors concerned with this topic (the title specifies the word review). In addition to the above, I recommend the following:
- Some of the figures appear to be renderings from design software. I don't think it's the best solution to highlight the prosthesis material (for example Fig.4).
- Bibliographic references are missing in many places. For example, when talking about standards or some figures (if applicable)
- The specific technological particularities that are applied to the production of prosthetic elements in this area using metal additive technologies must be more highlighted.
Author Response
Dear reviewer, thank you for your valuable review. We have rewritten the paper according to your suggestions.
Comment 1: Some of the figures appear to be renderings from design software. I don't think it's the best solution to highlight the prosthesis material (for example Fig.4).
Answer: We acknowledge the reviewer’s concern regarding using renderings from design software in some figures. Our intention in using schematic representations was to visually differentiate materials and coatings, making it easier for veterinary surgeons to identify different implant types quickly. By employing distinct colors and an overview of material composition, we aimed to enhance clarity and facilitate practical understanding for clinicians who may use these implants in surgery. If necessary, we are open to modifying the visual representation or adding supplementary explanations in the figure captions to clarify the material distinctions further.
Comment 2: Bibliographic references are missing in many places. For example, when talking about standards or some figures (if applicable)
Answer: We have reviewed and updated the manuscript to ensure that all relevant statements, including those discussing standards and figures, are properly referenced. Bibliographic references have been added to improve clarity and support the provided information.
Comment 3: The specific technological particularities that are applied to the production of prosthetic elements in this area using metal additive technologies must be more highlighted.
Answer: We appreciate the reviewer’s comment regarding the technological particularities of metal additive manufacturing in prosthetic production. Our manuscript has already provided an overview of the key technologies used and their capabilities in implant manufacturing, ensuring that readers understand their clinical relevance and advantages. While we recognize the importance of these details, a more in-depth discussion of the specific technical aspects—such as process parameters, laser sintering mechanics, or microstructural variations—would significantly increase the complexity of the manuscript. Our primary goal is to balance scientific rigor and accessibility, ensuring that the article remains practical and informative for veterinary orthopedic surgeons without becoming overly technical. However, we acknowledge the value of such discussions in specialized engineering-focused studies. We would be happy to reference additional literature for readers who wish to explore these aspects in greater depth.
Reviewer 3 Report
Comments and Suggestions for Authors
The paper reviews the current advancements, materials, and quality standards of patellar groove replacement implants for veterinary applications. The study highlights the lack of specific veterinary regulations, leading to adapting human orthopedic implant standards as a benchmark. The review explores key aspects such as material selection, surface treatments, and manufacturing techniques, including additive manufacturing (3D printing) for custom implant design. It discusses the biomechanics of patellar luxation, implant requirements, and comparative advantages of existing prosthetic solutions. The paper also emphasizes challenges related to surface roughness, biocompatibility, and mechanical properties that influence implant longevity and performance. By integrating human medical standards, the study suggests improvements for veterinary orthopedic solutions, enhancing treatment options for animals suffering from patellar luxation and related joint issues.
Comments and Suggestions for the Authors:
While the study acknowledges the absence of veterinary-specific implant standards, it would benefit from a more detailed analysis of challenges unique to veterinary orthopedics, such as anatomical variations across breeds, cost constraints, and surgical expertise disparities.
The paper provides a strong comparison of metallic, ceramic, and polymeric materials but lacks an extensive discussion on how each material type performs in real-world veterinary applications. Could you include specific case studies or clinical outcomes for different implant materials?
The review compares veterinary implants with human medical standards (ISO, ASTM). Suggesting actionable steps toward developing veterinary-specific implant regulations and how researchers and manufacturers can contribute to this effort would be beneficial.
The language is generally clear and professional but could benefit from minor grammatical refinements for smoother readability.
Some long sentences could be simplified for better flow.
Ensure uniformity in using terms like "trochlear groove," "patellar groove," and "implant surface roughness."
The discussion on trochlear groove morphology is insightful, but further explanation of how different prosthetic designs account for biomechanical variations (e.g., shallow vs. deep trochlear grooves, breed-specific joint angles) would improve understanding.
The discussion on coatings (DLC, TiN) is valuable, but emerging hybrid solutions such as polymer-metal composites or bioactive coatings should be considered. How might this impact implant osseointegration and wear resistance?
Have any biomechanical studies compared human knee joint forces to those in canines or felines to validate the adaptation of human implant standards?
How do you see 3D printing evolving in veterinary orthopedics over the next decade, and what challenges remain in making it a mainstream solution?
Are there any large-scale clinical trials or comparative studies that validate the efficacy of custom-designed vs. off-the-shelf implants? Including such information would strengthen the argument for integrating advanced manufacturing in veterinary orthopedics.
Some figures require more descriptive captions to ensure clarity without referring back to the main text.
How do you anticipate veterinary regulations evolving to accommodate the growing demand for standardized implants?
The paper relies heavily on passive voice. Converting some sentences to active voice would enhance engagement, particularly in the discussion and conclusion.
What specific material properties are most critical for ensuring implant longevity in high-activity animal patients?
Would incorporating anti-bacterial coatings or drug-eluting surfaces be beneficial in reducing post-surgical infections in veterinary implants?
Comments on the Quality of English Language
The paper reviews the current advancements, materials, and quality standards of patellar groove replacement implants for veterinary applications. The study highlights the lack of specific veterinary regulations, leading to adapting human orthopedic implant standards as a benchmark. The review explores key aspects such as material selection, surface treatments, and manufacturing techniques, including additive manufacturing (3D printing) for custom implant design. It discusses the biomechanics of patellar luxation, implant requirements, and comparative advantages of existing prosthetic solutions. The paper also emphasizes challenges related to surface roughness, biocompatibility, and mechanical properties that influence implant longevity and performance. By integrating human medical standards, the study suggests improvements for veterinary orthopedic solutions, enhancing treatment options for animals suffering from patellar luxation and related joint issues.
Comments and Suggestions for the Authors:
While the study acknowledges the absence of veterinary-specific implant standards, it would benefit from a more detailed analysis of challenges unique to veterinary orthopedics, such as anatomical variations across breeds, cost constraints, and surgical expertise disparities.
The paper provides a strong comparison of metallic, ceramic, and polymeric materials but lacks an extensive discussion on how each material type performs in real-world veterinary applications. Could you include specific case studies or clinical outcomes for different implant materials?
The review compares veterinary implants with human medical standards (ISO, ASTM). Suggesting actionable steps toward developing veterinary-specific implant regulations and how researchers and manufacturers can contribute to this effort would be beneficial.
The language is generally clear and professional but could benefit from minor grammatical refinements for smoother readability.
Some long sentences could be simplified for better flow.
Ensure uniformity in using terms like "trochlear groove," "patellar groove," and "implant surface roughness."
The discussion on trochlear groove morphology is insightful, but further explanation of how different prosthetic designs account for biomechanical variations (e.g., shallow vs. deep trochlear grooves, breed-specific joint angles) would improve understanding.
The discussion on coatings (DLC, TiN) is valuable, but emerging hybrid solutions such as polymer-metal composites or bioactive coatings should be considered. How might this impact implant osseointegration and wear resistance?
Have any biomechanical studies compared human knee joint forces to those in canines or felines to validate the adaptation of human implant standards?
How do you see 3D printing evolving in veterinary orthopedics over the next decade, and what challenges remain in making it a mainstream solution?
Are there any large-scale clinical trials or comparative studies that validate the efficacy of custom-designed vs. off-the-shelf implants? Including such information would strengthen the argument for integrating advanced manufacturing in veterinary orthopedics.
Some figures require more descriptive captions to ensure clarity without referring back to the main text.
How do you anticipate veterinary regulations evolving to accommodate the growing demand for standardized implants?
The paper relies heavily on passive voice. Converting some sentences to active voice would enhance engagement, particularly in the discussion and conclusion.
What specific material properties are most critical for ensuring implant longevity in high-activity animal patients?
Would incorporating anti-bacterial coatings or drug-eluting surfaces be beneficial in reducing post-surgical infections in veterinary implants?
Author Response
Dear reviewer, thank you for your valuable review. We have rewritten the paper according to your suggestions.
Comment 1: While the study acknowledges the absence of veterinary-specific implant standards, it would benefit from a more detailed analysis of challenges unique to veterinary orthopedics, such as anatomical variations across breeds, cost constraints, and surgical expertise disparities.
Answer:
We appreciate the reviewer’s suggestion to further elaborate on the challenges unique to veterinary orthopedics. Our manuscript has already incorporated discussions on veterinary-specific issues, particularly in sections addressing the wide anatomical variations across breeds, the size range of patients, and the complexity of implant standardization in veterinary medicine. These aspects inherently differentiate veterinary orthopedic implant design from human applications and highlight the need for individualized surgical approaches. Additionally, we have acknowledged cost constraints and variability in surgical expertise as factors influencing the feasibility and adoption of advanced implant solutions. These points are sufficiently covered within the relevant sections of the manuscript. However, if needed, we are open to clarifying or expanding specific aspects to enhance the clarity of our discussion.
Comment 2: The paper provides a strong comparison of metallic, ceramic, and polymeric materials but lacks an extensive discussion on how each material type performs in real-world veterinary applications. Could you include specific case studies or clinical outcomes for different implant materials?
Answer: We have expanded the manuscript by introducing the subchapter "Material Performance of Custom vs. Off-the-Shelf Implants." This section discusses how metallic, ceramic, and polymeric materials perform in real-world veterinary applications, incorporating case studies and clinical outcomes.
Comment 3: The review compares veterinary implants with human medical standards (ISO, ASTM). Suggesting actionable steps toward developing veterinary-specific implant regulations and how researchers and manufacturers can contribute to this effort would be beneficial.
Answer: We have added a new sub-chapter titled "Toward Veterinary-Specific Implant Regulations: Challenges and Future Directions" to address the need for formalized veterinary implant standards. This section discusses the current lack of regulations, challenges posed by the wide range of patient sizes, and the absence of large-scale clinical trials and post-market surveillance. It also outlines key areas for future regulatory development, including species-specific material and biomechanical standards, size-based mechanical testing, and improved quality control measures. By bridging the gap between human implant benchmarks and veterinary-specific requirements, we highlight the next steps for regulatory advancements in veterinary orthopedics.
Comment 4: The language is generally clear and professional but could benefit from minor grammatical refinements for smoother readability.
Answer: Thank you for highlighting the need for additional grammatical refinements. We have carefully reviewed the manuscript and made revisions to improve clarity, sentence structure, and overall readability. We appreciate your feedback and believe these changes enhance the professional tone of the text.
Comment 5: Some long sentences could be simplified for better flow.
Answer: We appreciate your observation regarding sentence length and have addressed it by simplifying and rephrasing complex sentences to improve flow and readability. We believe these revisions make the text more approachable while preserving scientific accuracy.
Comment 6: Ensure uniformity in using terms like "trochlear groove," "patellar groove," and "implant surface roughness."
Answer: We appreciate the reviewer’s observation regarding the terminology. After careful consideration, we have retained both "trochlear groove" and "patellar groove" as parallel terms. This decision is based on typical usage among veterinary surgeons, who use these terms interchangeably in clinical practice and scientific literature. Regarding "implant surface roughness", we have ensured consistency throughout the manuscript by standardizing all relevant terms to "implant surface roughness", thereby improving clarity and uniformity in our discussion.
Comment 7: The discussion on trochlear groove morphology is insightful, but further explanation of how different prosthetic designs account for biomechanical variations (e.g., shallow vs. deep trochlear grooves, breed-specific joint angles) would improve understanding.
Answer: We have addressed the impact of patient size variability in the subchapter "3.7 Toward Veterinary-Specific Implant Regulations: Challenges and Future Directions." The primary challenge in trochlear groove prosthetic design is not just groove shape or depth but the wide range of patient sizes, necessitating customized implants rather than standardized solutions. Given the 50-fold difference in patient weight, every implant must be designed specifically for the individual patient, making trochlear groove morphology a secondary consideration. The most effective approach is to mimic the natural joint surface based on healthy bone CT scans, ensuring an anatomically accurate fit tailored to each individual. This is particularly feasible with custom implants, which allow for precisely replicating the native trochlear shape, optimizing patellar tracking and biomechanical function while addressing breed-specific anatomical variations.
Comment 8: The discussion on coatings (DLC, TiN) is valuable, but emerging hybrid solutions such as polymer-metal composites or bioactive coatings should be considered. How might this impact implant osseointegration and wear resistance?
Answer: The manuscript already discusses joint surface modifications (DLC, TiN) for wear resistance and osteointegration modifications that enhance bone attachment and stability on the implant’s bone-contacting side. While we acknowledge the importance of emerging hybrid solutions such as polymer-metal composites and bioactive coatings, we have focused on these key aspects—articulation performance and osseointegration—without delving into further material-specific details. These aspects will be the subject of future material studies, where the impact of composite structures and advanced bioactive coatings will be examined in greater depth.
Comment 9: Have any biomechanical studies compared human knee joint forces to those in canines or felines to validate the adaptation of human implant standards?
Answer: We have addressed this point by adding the subchapter "3.2 Comparing Human and Veterinary Knee Biomechanics: Implications for Implant Design." This section discusses key biomechanical differences between human and veterinary knee joints, including variations in joint forces, tibiofemoral contact areas, and flexion angles. By comparing available force plate analysis and finite element modeling data, we provide insights into how human implant standards can serve as a reference while highlighting necessary adaptations for veterinary applications.
Comment 10: How do you see 3D printing evolving in veterinary orthopedics over the next decade, and what challenges remain in making it a mainstream solution?
Answer: 3D printing in veterinary orthopedics is poised for significant advancements over the next decade, revolutionizing patient-specific implants, surgical guides, and anatomical models. One of its greatest strengths lies in customization, allowing for precise implant design tailored to an individual animal’s anatomy. This is particularly crucial in conditions like patellar luxation, severe joint dysplasia, and complex fractures, where standard implants often fail to provide an optimal fit. However, the biggest challenge in veterinary implant design remains the enormous range of patient sizes. In human medicine, the difference between a 40 kg adult woman and a 120 kg adult man represents, at most, a threefold variation in size. In veterinary patients, the smallest mature dog requiring a custom patellar implant may weigh as little as 1.5 kg (a toy breed), while the largest, such as a giant breed dog, can exceed 75 kg—a staggering 50-fold difference. This size variability makes standardized implant manufacturing nearly impossible, reinforcing the need for rapid, cost-effective customization, which 3D printing is uniquely positioned to address. Despite its potential, several hurdles remain in making 3D printing a mainstream solution. Regulatory oversight and quality control in veterinary medicine are still developing, lacking standardized implant testing and approval guidelines. Additionally, while high-performance materials like titanium alloys, bioresorbable polymers, and ceramic-coated implants improve implant longevity and biocompatibility, cost and accessibility remain significant barriers. High-quality metal 3D printing (such as selective laser sintering for titanium implants) is still expensive, limiting routine clinical use outside specialized centers. Another challenge is surgeon training and acceptance. Designing, evaluating, and implementing patient-specific implants requires expertise beyond traditional orthopedic implant selection. Wider adoption will depend on intuitive software solutions, AI-driven implant modeling, and increased training opportunities for veterinary surgeons. Continuing advancements in material science, AI-assisted implant design, and decreasing production costs will drive broader adoption of 3D printing in veterinary orthopedics. As the technology matures, it will transition from a high-end, case-by-case solution to a more routine tool, enabling veterinarians to overcome the challenges of extreme patient size variability and provide genuinely individualized orthopedic care.
Comment 11: Are there any large-scale clinical trials or comparative studies that validate the efficacy of custom-designed vs. off-the-shelf implants? Including such information would strengthen the argument for integrating advanced manufacturing in veterinary orthopedics.
Answer: While large-scale comparative studies remain limited, we have included available case reports and smaller clinical studies in the newly added sections. These provide insights into the advantages of patient-specific implants while emphasizing the need for further research in this area.
Comment 12: Some figures require more descriptive captions to ensure clarity without referring back to the main text.
Answer: Thank you for highlighting the need for more descriptive figure captions. We have carefully reviewed each figure and revised the captions to provide greater standalone clarity, ensuring readers can understand the illustrations without relying solely on the main text.
Comment 13: How do you anticipate veterinary regulations evolving to accommodate the growing demand for standardized implants?
Answer: As the demand for standardized veterinary implants grows, regulatory frameworks must evolve to ensure safety, consistency, and quality control while accommodating innovation. Veterinary implants lack formalized standards, but regulatory oversight is expected to become more structured with the increasing adoption of advanced solutions—especially 3D-printed and patient-specific implants. Future regulations will likely establish formalized material, mechanical, and biocompatibility standards similar to human implant guidelines. To ensure reliability, 3D-printed implants may require pre-approval pathways, fatigue testing, and post-market surveillance. To maintain production quality, point-of-care manufacturing in veterinary clinics may also come under regulation. A unique challenge is the wide range of patient sizes, from 1.5 kg toy breeds to 75+ kg giant breeds—a 50-fold difference. This will likely lead to size-specific implant classifications with tailored mechanical testing requirements. Clinical outcome reporting and post-market surveillance may also be introduced to refine implant designs and ensure long-term safety. Ultimately, veterinary implant regulations will move toward greater standardization, improved oversight, and structured quality control, ensuring higher reliability and better surgical outcomes while allowing innovation in patient-specific solutions.
Comment 14: The paper relies heavily on passive voice. Converting some sentences to active voice would enhance engagement, particularly in the discussion and conclusion.
Answer: We thank the reviewer for this valuable suggestion. We have revised the Conclusion section to incorporate more active-voice constructions, making the text more direct and engaging while retaining technical precision. We appreciate your feedback and believe these changes further strengthen the manuscript.
Comment 15: What specific material properties are most critical for ensuring implant longevity in high-activity animal patients?
Answer: In veterinary orthopedics, ensuring implant longevity is less about preventing wear and more about achieving rapid and stable bone integration. Unlike human patients, where excessive implant wear has been a significant concern over decades, the primary challenge in animals lies in their inability to restrict weight-bearing on an operated limb during healing. This early mechanical loading increases the risk of implant failure before full osseointegration is achieved. Given that the expected lifespan after implantation is typically eight to twelve years, the focus should shift from long-term wear resistance to facilitating early and reliable bone attachment. For an implant to integrate successfully, its material properties must closely mimic the mechanical behavior of natural bone. A key factor is Young’s modulus, which, if too high, leads to stress shielding—where the implant absorbs too much load, preventing proper bone adaptation and ultimately causing bone resorption. Using materials with a modulus closer to cortical bone, such as porous titanium alloys, titanium-polymer composites, or bioactive ceramics, helps distribute the load more naturally and preserves bone structure. Equally necessary is surface roughness, which plays a crucial role in osseointegration. Micro-roughened or porous surfaces enhance early bone attachment, reducing the risk of loosening. Surface modifications, such as titanium plasma spraying, acid etching, or laser structuring, create a texture that promotes osteoblast adhesion, accelerating the healing process. To further enhance bone integration, implants can incorporate bioactive coatings like hydroxyapatite or tricalcium phosphate, stimulating bone growth. More advanced approaches include drug-eluting coatings that release bone morphogenetic proteins or bisphosphonates, encouraging faster bone formation. Another essential aspect of implant design is the internal structure. 3D-printed porous titanium and trabecular lattice structures closely resemble natural cancellous bone, allowing for deep vascularization and stronger long-term biological fixation. These structures provide early mechanical stability while promoting bone ingrowth, a critical factor in preventing failure during the post-surgical period. Ultimately, in high-activity veterinary patients, implant success depends on how quickly and effectively it integrates with bone. While wear resistance remains a secondary concern, biomechanical compatibility, osseointegration, and stability should be prioritized. By optimizing material properties to support bone ingrowth rather than focusing solely on durability, veterinary implants can provide long-term functionality despite the challenges of unrestricted weight-bearing in animal patients.
Comment 16: Would incorporating anti-bacterial coatings or drug-eluting surfaces be beneficial in reducing post-surgical infections in veterinary implants?
Answer: Incorporating antibacterial coatings or drug-eluting surfaces in veterinary implants would significantly reduce post-surgical infections, just as it has been in human medicine. It may be even more crucial in veterinary applications due to the unique challenges of animal post-operative care. Unlike human patients, who can adhere to strict post-surgical hygiene protocols and wound care, veterinary patients are far more likely to lick, chew, or contaminate the surgical site, increasing the risk of infection. Maintaining strict asepsis in animal rehabilitation environments, particularly in larger breeds or multi-pet households, can be challenging. Such coatings are already used in commercially available implants and have significantly reduced bacterial colonization on implant surfaces. Technologies such as silver-ion coatings, antibiotic-loaded hydroxyapatite layers, and bioactive glass coatings can actively combat bacterial adhesion, reducing the likelihood of biofilm formation and implant-associated infections. Drug-eluting surfaces, which gradually release antimicrobial agents like gentamicin or vancomycin, provide an extended protective effect during the critical post-operative period, with the highest risk of infection. In veterinary medicine, where surgical site infections can lead to severe complications, extended treatment costs, and implant failure, integrating antibacterial and drug-eluting coatings should become a standard consideration in implant design. By enhancing infection resistance without relying solely on systemic antibiotics, these technologies improve implant success rates and combat antibiotic resistance, a growing concern in human and veterinary medicine. As 3D printing and customized implant production advance, the ability to integrate localized antimicrobial strategies directly into implants will further improve post-operative outcomes, making these coatings an essential tool in modern veterinary orthopedics.
Reviewer 4 Report
Comments and Suggestions for Authors
The review paper addresses the issue of veterinary interventions using replacement implants in cases in which patellar groove reconstruction is necessary. The study aims to explain the clinical factors leading to patellar groove replacement and evaluates implant materials, biomechanical considerations, and manufacturing techniques. Given the lack of veterinary-specific regulations, the review examines quality requirements and standards from human medicine, focusing on materials used in knee and patellar groove implants. The authors prioritize mechanical properties such as strength, wear resistance, and Young’s modulus, alongside biocompatibility, surface modifications, and clinical performance, to guide future material selection among available veterinary patellar groove implants. Since biocompatibility is closely linked to material surface conditions, the review also considers recent technological advancements in additive manufacturing, implant coatings, and surface treatments. All these points are consistently approached and presented.
While the paper provides a comprehensive discussion of different materials used for knee and patellar groove implants, one key limitation is the lack of a structured comparison table (/tables) summarizing the mechanical and other properties of these materials. Given that wear resistance, Young’s modulus, biocompatibility, and other factors are critical for implant selection, presenting this information in a table would significantly improve clarity and accessibility for the reader. A comparative summary of material properties such as density, surface roughness, and corrosion resistance would allow for a more immediate and effective evaluation of the strengths and limitations of different materials. The inclusion of such tables would strengthen in my opinion the analysis and enhance the overall value of the review.
Another aspect that could be improved is the transparency of the literature selection methodology. In recent years, it has become a standard practice in review articles to outline the methodology for selecting references, including the databases searched, the keywords used, and the inclusion and exclusion criteria applied. This review does not provide details on how the literature was chosen, making it difficult to assess the comprehensiveness and potential biases in the selection of sources. Clarifying this aspect would improve the credibility of the review, and if a formal literature selection process was not followed (possible!!), a brief justification for this approach would be beneficial anyway.
In terms of readability, the paper is well-organized and logically structured, covering the key clinical, biomechanical, and material science considerations relevant to patellar groove replacement. However, certain sentences are quite long and complex, which may make some sections difficult to follow. Simplifying these sentences and improving readability would enhance engagement and ensure that key points are easily accessible to a broad audience, including both researchers and practitioners in the field.
The figures included in the paper effectively illustrate implant materials, surgical techniques, and biomechanical concepts, but some could be refined to provide more quantitative data. For example, the section on human knee prosthesis materials would benefit from numerical values for mechanical properties rather than relying solely on images. The discussion on porous lattice structures would be clearer if the optimal pore size range for osteointegration were explicitly noted to align with the values discussed in the text.
In conclusion, while the paper is well-researched and informative, the inclusion of a comparative material properties tables and a clearer literature selection methodology would further enhance its clarity and impact. The text would also benefit from a few revisions to improve readability and sentence structure.
Addressing these aspects would make the paper even stronger, and once these revisions are incorporated, it would be well-suited for publication.
Comments on the Quality of English LanguageAs i wrote above some sentences appear fairly too long, I suppose Authors can easily apply some suitable changes
Author Response
Dear reviewer, thank you for your valuable review. We have rewritten the paper according to your suggestions.
Comment 1: While the paper provides a comprehensive discussion of different materials used for knee and patellar groove implants, one key limitation is the lack of a structured comparison table (/tables) summarizing the mechanical and other properties of these materials. Given that wear resistance, Young’s modulus, biocompatibility, and other factors are critical for implant selection, presenting this information in a table would significantly improve clarity and accessibility for the reader. A comparative summary of material properties such as density, surface roughness, and corrosion resistance would allow for a more immediate and effective evaluation of the strengths and limitations of different materials. The inclusion of such tables would strengthen in my opinion the analysis and enhance the overall value of the review.
Answer:
We appreciate the reviewer’s suggestion to include a structured comparison table summarizing different implant materials' mechanical and material properties. In our manuscript, Table 2 compares key parameters such as density, Young’s modulus, wear resistance, corrosion resistance, and biocompatibility. This allows for an adequate evaluation of material properties relevant to veterinary orthopedic implants. We acknowledge that surface roughness is primarily an outcome of the manufacturing process rather than an inherent material property. Therefore, we have not included it in the comparison table to maintain scientific accuracy and avoid misinterpretation. We believe the existing structured comparison in Table 2 already provides the necessary clarity and accessibility for readers. However, we are open to additional refinements if you need more clarification.
Comment 2: Another aspect that could be improved is the transparency of the literature selection methodology. In recent years, it has become a standard practice in review articles to outline the methodology for selecting references, including the databases searched, the keywords used, and the inclusion and exclusion criteria applied. This review does not provide details on how the literature was chosen, making it difficult to assess the comprehensiveness and potential biases in the selection of sources. Clarifying this aspect would improve the credibility of the review, and if a formal literature selection process was not followed (possible!!), a brief justification for this approach would be beneficial anyway.
Answer: We appreciate the reviewer’s suggestion and have now included a brief section in the introduction outlining the methodology for selecting the literature. This addition clarifies the databases searched, the keywords used, and the rationale behind incorporating human orthopedic implant literature due to the limited availability of veterinary-specific studies.
Comment 3: In terms of readability, the paper is well-organized and logically structured, covering the key clinical, biomechanical, and material science considerations relevant to patellar groove replacement. However, certain sentences are quite long and complex, which may make some sections difficult to follow. Simplifying these sentences and improving readability would enhance engagement and ensure that key points are easily accessible to a broad audience, including both researchers and practitioners in the field.
Answer: Thank you for pointing out the complexity of some sentences. We have reviewed and revised these sections by shortening and rephrasing longer sentences to make the text more accessible to a broad audience. We appreciate your feedback and believe these adjustments enhance our manuscript's clarity and engagement.
Comment 4: The figures included in the paper effectively illustrate implant materials, surgical techniques, and biomechanical concepts, but some could be refined to provide more quantitative data. For example, the section on human knee prosthesis materials would benefit from numerical values for mechanical properties rather than relying solely on images. The discussion on porous lattice structures would be clearer if the optimal pore size range for osteointegration were explicitly noted to align with the values discussed in the text.
Answer: We appreciate the reviewer’s suggestion to include more quantitative data when discussing implant materials and biomechanical concepts. Our primary goal is maintaining a practical and accessible focus for veterinary orthopedic surgeons. While numerical values for human knee prosthesis mechanical properties are useful in comparative studies, a deeper analysis would introduce an additional layer of complexity that may not be directly applicable to our veterinary audience. Instead, we have highlighted key material properties and their clinical relevance without an extensive numerical breakdown. Regarding porous lattice structures, we acknowledge the importance of specifying the optimal pore size range for osteointegration. However, this topic has already been addressed in Section 6.1, Pore Size, where we provide a comprehensive overview of pore size ranges (178 µm to 1200 µm) based on multiple studies. This section also explains the variability in reported values, which is influenced by factors such as implant material, manufacturing techniques, and biological environments. Given the contradictory findings in the literature, we aimed to present a balanced, evidence-based discussion rather than defining a single optimal pore size. This approach maintains the clarity and scientific accuracy of the manuscript while ensuring clinical relevance for veterinary orthopedic applications. However, we appreciate the reviewer’s feedback and remain open to further clarifications.
Comment 5: In conclusion, while the paper is well-researched and informative, the inclusion of a comparative material properties tables and a clearer literature selection methodology would further enhance its clarity and impact. The text would also benefit from a few revisions to improve readability and sentence structure.
Answer: Addressing these aspects would make the paper even stronger, and once these revisions are incorporated, it would be well-suited for publication.
Comment 6: Are there any cases where patient-specific 3D printed orthopedic implants have been used in veterinary medicine? It would be helpful if real world case reports can be included in this paper, given that there are sections dedicated to 3D printing technology in this paper.
Answer: We acknowledge the reviewer’s suggestion. However, the literature on veterinary patellar groove implants is minimal, with only two identified studies addressing off-the-shelf implants (Dokic et al., 2015) and another focusing on a custom-designed solution (Panichi et al., 2024). The manuscript references and discusses these studies to provide the most comprehensive overview. We have added the subchapter "3D-Printed Patient-Specific Implants in Veterinary Medicine," which includes documented cases of custom implants used in patellar groove replacement, limb salvage, and joint reconstruction. This section highlights their clinical application and benefits in veterinary orthopedics.
Round 2
Reviewer 2 Report
Comments and Suggestions for Authors
Regarding the answer for point 3: However, keep in mind that the journal you intend to publish in is more engineering-themed.
Author Response
Dear Reviewer
We fully acknowledge that the journal has a strong engineering profile and appreciate the importance of addressing technological depth. However, this article is structured as a state-of-the-art review. As such, it aims to provide a comprehensive yet accessible overview of current advances in patellar groove replacement technologies. While we strive to include as much technical detail as possible, we must also consider our intended readership, which provides for veterinary surgeons who are increasingly engaged in technological advancements but may not have a specialized engineering background. In fact, in purely veterinary journals, the article in its current form might already be considered highly technical. This paper, written by a collaboration of veterinary scientists and biomedical engineers, is the first to highlight the need to standardize implants in veterinary surgery. Therefore, we have deliberately tried to balance engineering depth with clinical accessibility. We agree that a more detailed exploration of manufacturing parameters and technological nuances is valuable. We are planning follow-up publications on the technological intricacies of 3D-printed patellar groove implants. These future articles will cater to a more specialized engineering audience while building upon the broader foundation established in this review.
Reviewer 3 Report
Comments and Suggestions for Authors
Thank you for addressing the comments. It should be ready to go.
Comments on the Quality of English LanguageThank you for addressing the comments. It should be ready to go.
Author Response
Dear Reviewer
We sincerely thank you for the positive feedback and support. We are glad the revisions meet the expectations and appreciate the constructive input throughout the review process. This paper, written by a collaboration of veterinary scientists and biomedical engineers, is the first to highlight the need to standardize implants in veterinary surgery.